# Effect of stress history on sediment transport and channel adjustment in graded gravel-bed rivers

Chenge An[1,2], Marwan A. Hassan[2], Carles Ferrer-Boix[3], Xudong Fu[1]

[1] Department of Hydraulic Engineering, State Key Laboratory of Hydroscience and Engineering, Tsinghua University, Beijing, China.

[2] Department of Geography, The University of British Columbia, Vancouver, BC, Canada.

[3] Serra Húnter Fellow, Department of Graphic and Design Engineering, Technical University of Catalonia, Barcelona, Spain.

*Correspondence to*: Xudong Fu (xdfu@tsinghua.edu.cn)

**Abstract.** With the increasing attention on environmental flow management for the maintenance of habitat diversity and ecosystem health of mountain gravel-bed rivers, much interest has been paid to how inter-flood low flow can affect gravel-bed river morphodynamics during subsequent flood events. Previous research has found that antecedent conditioning flow can lead to an increase in the critical shear stress and a reduction in sediment transport rate during a subsequent flood. But how long this effect can last during the flood event has not been fully discussed. In this paper, a series of flume experiments with various durations of conditioning flow are presented to study this problem. Results show that channel morphology adjusts significantly within the first 15 minutes of the conditioning flow, but becomes rather stable during the remainder of the conditioning flow. The implementation of conditioning flow can indeed lead to a reduction of sediment transport rate during the subsequent hydrograph, but such effect is limited only within a relatively short time at the beginning of the hydrograph. This indicates that bed reorganization during the conditioning phase, which induce the stress history effect, is likely to be erased with increasing intensity of flow and sediment transport during the subsequent flood event.

## 1 Introduction

Prediction of sediment transport is of vital importance because it is related to many aspects of river dynamics and management, including river morphodynamics modeling (Parker, 2004), river restoration (Chin et al., 2009), aquatic habitats (Montgomery et al., 1996), natural hazard planning (Marston, 2008), bedrock erosion (Sklar and Dietrich, 2004), and landscape evolution (Howard, 1994). In gravel-bed rivers, sediment transport is controlled by flow magnitude and flashiness, sediment supply, bed surface structures, channel morphology and the grain size distribution (GSD) of sediment (Montgomery and Buffington, 1997; Masteller et al., 2019). Therefore, prediction of sediment transport in mountain rivers still remains difficult despite the large body of existing theories. This is due to the fact that these theories were mostly developed for lowland streams with continuous sediment supply and an average flow regime, which do not apply to mountain streams (Gomez and Church, 1989; Rickenmann, 2001; Schneider et al., 2015).

For example, the hydrograph of mountain gravel-bed rivers is often characterized by large fluctuations of flow discharge, including both short-term flash flood and long-term inter-flood low flow (Powell et al., 1999). However, research on the morphodynamics of mountain rivers often focuses on the effects of floods (or constant high flow) and neglects the role of inter-flood low flow, with the consideration that most sediment transport and morphological adjustments of mountain rivers occur during relatively high flows (Klingeman and Emmett, 1982; Paola et al., 1992).

Reid and colleagues (Reid and Frostick, 1984; Reid et al., 1985) studied the effects of inter-flood low flow on subsequent sediment transport in Turkey Brook, England. They found that bedload transport rates were reduced during relatively isolated flood events (e.g., events separated by long time intervals) compared to those that were closely spaced, with the entrainment threshold up to as large as three times higher. They linked this with sediment reorganization during prolonged periods of antecedent flow, which can make the river bed more armored and more resistant to entrainment, thus delaying the onset of sediment mobility in the following flood event. Carling et al. (1992) also reported differences in the initial motion criteria between flood events due to changes in packing and orientation of sediment particles.

To further study such "memory" effects of antecedent flow on the sediment transport during a subsequent flood, a number of flume experiments as well as field surveys have been conducted in the past decade, and different terms have been proposed, including "stress history effect" (Monteith and Pender, 2005; Paphitis and Collins, 2005; Haynes and Pender, 2007; Ockelford and Haynes, 2013), "flood history effect" (Mao, 2018), "flow history" (Masteller et al., 2019). The difference in the terminology could be partly due to the available data and the chosen approach in different research works. Here we adopt the term "stress history" in this paper. It should also be noted that the approach based on shear stress (and therefore terminology), even though widely applied for laboratory experiments, is much less reliable for field measurements.

Paphitis and Collins (2005) conducted flume experiments to study the entrainment threshold of uniform sediment subjected to antecedent flow durations of up to 120 minutes. They found that with a longer and higher antecedent flow, the critical bed shear stress increases and the total bedload flux decreases. The work of Paphitis and Collins (2005) was extended by Monteith and Pender (2005) and Haynes and Pender (2007) to consider bimodal sand-gravel mixtures. They found that for a graded bed, longer periods of antecedent flow increase bed stability due to local particle rearrangement, in agreement with Paphitis and Collins (2005); whereas higher magnitudes of antecedent flow reduce bed stability due to selective entrainment of the fine matrix on bed surface, counter to Paphitis and Collins' (2005) conclusion based on uniform sediment. Haynes and Pender (2007) further analyzed the two competing effects and concluded that particle rearrangement may be of greater relative importance than the winnowing of the fine sediment as it affects subsequent sediment transport. By using high resolution laser scanning and statistical analysis of the bed topography, Ockelford and Haynes (2013) also demonstrated that the response of bed topography to stress history is grade specific: bed roughness decreased in uniform beds but increased in graded bed with an increase length of an antecedent flow period. Performing a series of flume experiments, Masteller and Finnegan (2017) studied the evolution of the river bed on particle scale during low flow. They linked reduction of bedload flux to the re-organization of the highest protruding grains (1%-5% of the entire bed) on bed surface.

Because of the above-mentioned research, existing sediment transport formulae for gravel-bed rivers (e.g. Meyer-Peter and Müller, 1948; Parker, 1990; Wilcock and Crowe, 2003; Wong and Parker, 2006) are regarded to be inaccurate because they do not take the effect of stress history into account. To this end, Paphitis and Collins (2005) proposed an empirical formula for the exposure correction factor in the critical shear velocity for a uniform sand-size bed based on their experimental data. Johnson (2016) developed a state function for the critical shear stress in terms of transport disequilibrium, which incorporates the effects of stress history and hydrograph variability. Ockelford et al. (2019) proposed two forms of functions to link the antecedent duration and the critical shear stress. The two alternatives proposed by Ockelford et al. (2019) correct the function proposed by Paphitis and Collins (2005), whose exposure correction uses a logarithmic function which implicitly assumes an unbound growth as antecedent time tends towards infinity.

Research to date has shown that antecedent flow can stabilize the river bed, thus influencing the threshold of sediment motion as well as bedload flux. However, most of the previous research about stress history is either under conditions with relatively low sediment transport or with relatively short durations of sediment transport in order to capture the threshold of sediment motion (Monteith and Pender, 2005; Paphitis and Collins, 2005; Haynes and Pender, 2007; Ockelford and Haynes, 2013; Masteller and Finnegan, 2017; Ockelford et al., 2019). On the other hand, other researchers have found that exceptionally high discharge events can reduce critical shear stress by disrupting particle interlocking and breaking of bed structure (Lenzi, 2001; Turowski et al., 2009; Turowski et al., 2011; Yager et al., 2012; Ferrer-Boix and Hassan 2015; Masteller et al., 2019). Flume experiments by Masteller and Finnegan (2017) also indicate an increase in the number of highly mobile, highly protruding grains in response to sediment transporting flows. Therefore, the effect of high discharge events in reducing the critical shear stress likely counterbalances the stress history effect of antecedent flow to increase the critical shear stress. Besides, the supply of fine sediment (during high discharge events) is also widely observed to enhance the mobilization of coarse sediment (Wilcock et al., 2001; Curran and Wilcock, 2005; Venditti et al., 2010). In consideration of these opposing mechanisms, how long can the stress history effect last during a subsequent flood event is not well understood. Such a question is important especially in light of the fact that most sediment transport and channel adjustment of mountain gravel-bed rivers occurs during high discharge events, when the flow shear stress is high.

In this paper, flume experiments consisting of high and low flow are conducted to study this problem. The experimental arrangement is described in Sect. 2. In Sect. 3, we present the experimental results showing how channel morphology and sediment transport during a subsequent hydrograph respond to various durations of antecedent conditioning flow. The threshold of motion is analyzed in Sect. 4 based on the experimental data. Implications and limitations of this study are also discussed in Sect. 4. Finally, conclusions are summarized in Sect. 5.

## 2 Experimental arrangements

The experimental arrangements were guided by conditions observed in East Creek, a small mountain creek in Malcom Knob Forest, University of British Columbia (for details on the study site see Papangelakis and Hassan, 2016). To investigate

the study objectives, we conducted flume experiments in the Mountain Channel Hydraulic Experimental Laboratory at the University of British Columbia. The experiments were conducted in a tilting flume with a length of 5 m, a width of 0.55 m and a depth of 0.80 m. The initial slope was 0.04 m/m. Water, but not sediment was recirculated by an axial pump. A set of six experiments (REF2 – REF7) was conducted; the experimental conditions are briefly summarized in Table 1. For experiments REF3 – REF7, the same hydrograph and sedimentograph were conducted, but with different durations of constant conditioning flow prior to the hydrograph/sedimentograph. It should be noted that in the experiments, we only implemented the rising limb of the hydrograph/sedimentograph, rather than a full hydrograph/sedimentograph with both rising and falling limbs. Rather than studying river adjustment during a flow hydrograph, we aimed at determining the influence of conditioning time on bedload and bed surface arrangements as flow rates increased. We denote these as REF3 (10), REF4 (2), REF5 (5), REF6 (15) and REF7 (0.25), with the numbers in the brackets denoting the duration of the conditioning flow in hours. Experiment REF2 (15) consists of a 15-hour conditioning period without a subsequent hydrograph/sedimentograph, to test the reproducibility of our experimental results during the conditioning flow.

**Table 1.** Summary of the experimental conditions and measurements. The experiments are listed in the table in order of decreasing duration of conditioning flow.

| No. | Phase | Duration (h) | Flow discharge (l/s) | Water surface slope (%) | Flow depth (cm) | Froude number (-) | $\tau_b$ (Pa) | $\Delta z_b$ (mm) | Sediment feed (kg/h) | $D_{s50}$ (mm) | $D_{s90}$ (mm) | $D_{l50}$ (mm) | $D_{l90}$ (mm) | $\tau^*_{s50}$ | $Q_s$ (kg/h) |
|---|---|---|---|---|---|---|---|---|---|---|---|---|---|---|---|
| REF2 (15) | Conditioning | 15 | 25 | 2.62 | 6.33 | 0.91 | 16.27 | -30.2 | 0 | 15.5 | 29.7 | 1.07 | 5.43 | 0.065 | 0.27 |
| REF6 (15) | Conditioning | 15 | 25 | 3.27 | 6.47 | 0.88 | 20.76 | -16.6 | 0 | 15.7 | 30.8 | 35.18 | 42.84 | 0.082 | 0.89 |
| | Step 1 | 2 | 26 | 3.34 | 6.39 | 0.94 | 20.93 | 0.3 | 1 | 14.4 | 30.0 | 12.51 | 39.38 | 0.090 | 0.68 |
| | Step 2 | 2 | 28 | 3.10 | 6.29 | 1.03 | 19.13 | 0.0 | 1.5 | 17.3 | 29.4 | 7.28 | 27.59 | 0.068 | 0.76 |
| | Step 3 | 2 | 32 | 3.06 | 6.80 | 1.05 | 20.41 | -1.9 | 3.2 | 16.2 | 31.8 | 12.39 | 36.54 | 0.078 | 6.73 |
| | Step 4 | 2 | 40 | 2.81 | 7.78 | 1.07 | 21.45 | -16.1 | 10 | 15.9 | 31.6 | 11.48 | 36.03 | 0.083 | 13.39 |
| REF3 (10) | Conditioning | 10 | 25 | 2.73 | 6.02 | 0.98 | 16.12 | -25.8 | 0 | 14.8 | 29.2 | 2.17 | 9.98 | 0.067 | 0.28 |
| | Step 1 | 2 | 26 | 2.75 | 5.93 | 1.04 | 16.00 | 0.1 | 1 | 15.6 | 29.5 | 2.55 | 19.94 | 0.063 | 1.71 |
| | Step 2 | 2 | 28 | 2.69 | 6.35 | 1.01 | 16.77 | 0.3 | 1.5 | 15.8 | 30.2 | 4.06 | 26.99 | 0.065 | 2.19 |
| | Step 3 | 2 | 32 | 2.88 | 6.81 | 1.04 | 19.25 | -1.7 | 3.2 | 15.9 | 30.1 | 6.18 | 24.26 | 0.075 | 2.44 |
| | Step 4 | 2 | 40 | 2.48 | 8.34 | 0.96 | 20.28 | -8.0 | 10 | 14.2 | 32.8 | 14.45 | 39.13 | 0.088 | 12.45 |
| REF5 (5) | Conditioning | 5 | 25 | 3.26 | 5.51 | 1.12 | 17.63 | -16.8 | 0 | 15.3 | 32.0 | 8.23 | 25.34 | 0.071 | 0.49 |
| | Step 1 | 2 | 26 | 3.24 | 6.19 | 0.98 | 19.68 | -0.6 | 1 | 15.4 | 31.5 | 6.57 | 23.63 | 0.079 | 2.24 |
| | Step 2 | 2 | 28 | 3.09 | 6.21 | 1.05 | 18.82 | -0.3 | 1.5 | 17.2 | 31.4 | 9.38 | 28.44 | 0.067 | 3.30 |
| | Step 3 | 2 | 32 | 3.05 | 6.65 | 1.08 | 19.91 | -1.2 | 3.2 | 16.8 | 31.9 | 11.90 | 47.91 | 0.073 | 5.72 |
| | Step 4 | 2 | 40 | 2.78 | 7.82 | 1.06 | 21.33 | -13.4 | 10 | 15.1 | 34.5 | 15.09 | 38.56 | 0.087 | 40.03 |
| REF4 (2) | Conditioning | 2 | 25 | 2.82 | 5.55 | 1.11 | 15.34 | -17.8 | 0 | 12.3 | 27.8 | 3.10 | 15.79 | 0.077 | 1.50 |
| | Step 1 | 2 | 26 | 2.73 | 5.55 | 1.16 | 14.85 | -0.5 | 1 | 14.8 | 28.9 | 3.90 | 20.31 | 0.062 | 0.96 |
| | Step 2 | 2 | 28 | 2.71 | 6.19 | 1.06 | 16.46 | -0.1 | 1.5 | 15.6 | 29.2 | 6.28 | 46.76 | 0.065 | 2.41 |
| | Step 3 | 2 | 32 | 3.15 | 6.85 | 1.04 | 21.15 | -6.4 | 3.2 | 14.5 | 28.8 | 17.34 | 37.76 | 0.090 | 26.73 |
| | Step 4 | 2 | 40 | 2.76 | 8.01 | 1.02 | 21.69 | -7.7 | 10 | 13.7 | 29.7 | 10.88 | 35.45 | 0.098 | 5.23 |
| REF7 (0.25) | Conditioning | 0.25 | 25 | 3.46 | 6.20 | 0.94 | 21.06 | -14.9 | 0 | 14.0 | 29.5 | 10.54 | 28.03 | 0.093 | 19.44 |
| | Step 1 | 2 | 26 | 3.20 | 6.54 | 0.90 | 20.53 | -4.8 | 1 | 15.6 | 31.6 | 7.11 | 28.91 | 0.081 | 3.48 |
| | Step 2 | 2 | 28 | 3.14 | 6.58 | 0.96 | 20.27 | -0.7 | 1.5 | 16.2 | 21.2 | 6.91 | 30.73 | 0.077 | 2.52 |
| | Step 3 | 2 | 32 | 3.12 | 7.00 | 1.00 | 21.41 | -4.5 | 3.2 | 14.3 | 30.5 | 10.09 | 37.40 | 0.092 | 12.32 |
| | Step 4 | 2 | 40 | 2.73 | 8.29 | 0.97 | 22.19 | -9.6 | 10 | 17.3 | 33.6 | 12.13 | 30.78 | 0.079 | 16.80 |

a. $Q_s$: bedload transport rate, $\mathit{\Delta z_b}$: mean difference of bed elevation averaged over the whole river channel, $\tau_b$: shear stress, $D_{s50}$ and $D_{s90}$: $D_{50}$ and $D_{90}$ of bed surface,
$D_{l50}$ and $D_{l90}$: $D_{50}$ and $D_{90}$ of bedload, $\tau^*_{s50}$: Shields number for $D_{s50}$. Here $D_{90}$ denotes the grain size such that 90% is finer, and $D_{50}$ denotes the grain size such that
50% is finer. All values presented in this table are measured at the end of each stage, except for $\mathit{\Delta z_b}$ which denotes the mean difference of bed elevation during
each stage (i.e., difference between the end of this stage and the end of last stage). A positive value of $\mathit{\Delta z_b}$ denotes aggradation, and a negative value of $\mathit{\Delta z_b}$ denotes
degradation.

Figure 1 shows the water and sediment supply implemented during the experiments. The water discharge

was selected to represent typical flows in East Creek, with the 25 l/s flow during the conditioning period being
equivalent to half the bankfull flow, and the peak flow discharge of 40 l/s during the hydrograph being about 1.1 times
the bankfull flow in East Creek. Because the purpose of this paper is to study the evolution of bed stability, sediment
was not fed during the conditioning flow. For each step of the hydrograph, the feed rate of sediment was specified to
be close to the transport capacity of the flow. Determination of the sediment supply rates was facilitated by a numerical
model which was calibrated for similar experimental conditions (Ferrer-Boix and Hassan, 2014). Sediment was fed
into the flume at the upstream end using a conveyor belt feeder at the calculated transport rate capacity. The feed rate
of the sedimentograph ranged between 1 kg/h and 10 kg/h. Both the hydrograph and the sedimentograph consisted of
four steps, with each step lasting for 2 hours.

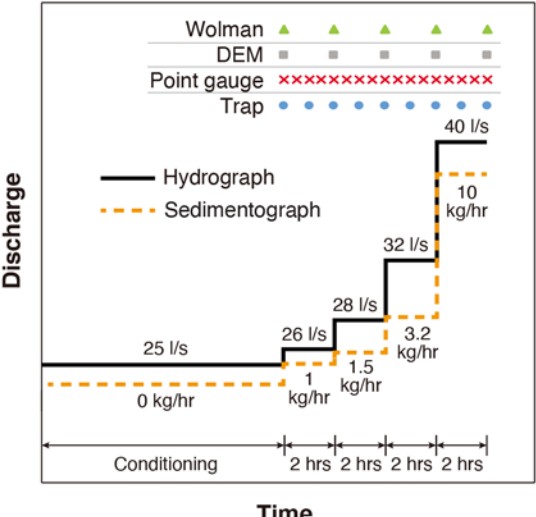

**Figure 1.** Water and sediment supply implemented in the experiments. Markers in top of the figure denote the time
of measurements during the hydrograph phase. Time of measurements during the conditioning phase is not shown in
this figure.

Figure 2 shows the GSD of the bulk sediment used in the experiments, with the grain size ranging between

0.5 and 64 mm. The GSD was scaled from East Creek by a ratio of 1:4, except that sediment (after scaling) with a
grain size less than 0.5 mm was excluded. This preserved the entire gravel distribution of East Creek with a maximum
size of 256 mm (scaled to 64 mm in Fig. 2). The model was "generic" rather than specific. This means that no attempt
was made to reproduce the geometric details of the prototype channel. The bulk sediment was sieved at half $\varphi$ intervals
and each grain size class was painted in different colors for texture analysis and visual identification. Before the
commencement of each experiment, we hand-mixed and leveled the bulk sediment to make a flat and uniform layer
of loose material with a depth of 0.15 m. The sediment was then slowly flooded and then drained to aid settlement.
The bulk sediment was also used for the sediment feed in each experiment.

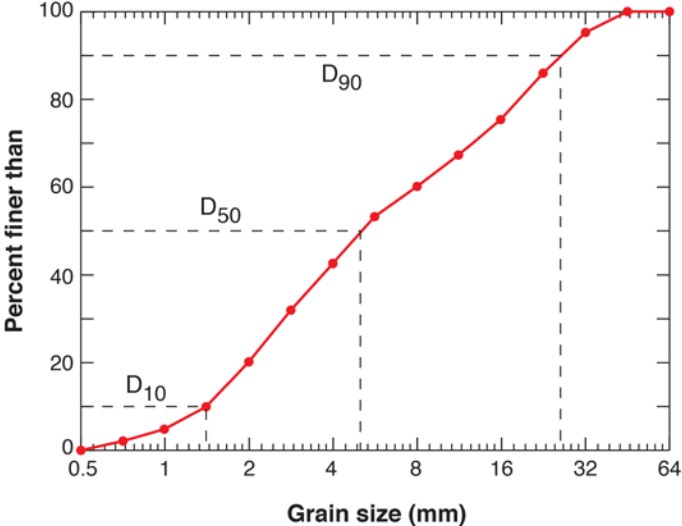

**Figure 2.** Grain size distribution of the bulk sediment used in the experiments.

The bed and water surface elevations were measured along the flume every 0.25 m using a mechanical point
gauge with a precision of ± 0.001 m. Water depth fluctuations due to wave effects at a point were about 5% or less.
Water surface slope and bed slope are calculated based on a linear regression of the point gauge data measured between
0.5 m and 4.75 m upstream of the outlet. The most upstream and downstream sections are excluded to avoid boundary
effects. A green laser scanner mounted on a motorized cart was also used to measure the bed surface elevation along
the flume. Bed laser scans were composed of cross sections spaced 2 mm apart with 1 mm vertical and horizontal
accuracy (for details see Elgueta-Astaburuaga and Hassan, 2017). The standard deviation of bed elevation was
calculated based on the DEM data from scans. Before the calculation of standard deviation, the DEM was detrended
based on linear regression to remove spatial trends with scales larger than the scale of sediment patterns (e.g., bed
slope or undulations). To estimate the particle size distribution of the bed surface we used digital cameras mounted
on a motorized cart along the entire flume. Images were merged together to visualize the bed and perform the particle
size analysis (Chartrand et al., 2018). To avoid distortion effects due to image merging, the width of the image strips
that were stitched to get a composite image was specified as just 2 cm. The particle size distribution of the bed surface
was estimated using the Wolman (point count) method, by identifying the grain size of particles at the intersections
of a 5 cm grid superimposed on the photograph. Individual grains were identified by color. For each experiment, the
grain size distribution of the bed surface was calculated at different times to quantify its changes during the experiment.
The sediment transport rates for various size ranges were measured at the end of the flume using a light table
(for details see Zimmerman et al., 2008; Elgueta-Astaburuaga and Hassan 2017) and automated image analysis at a
resolution of 1 second. Material evacuated from the flume was trapped in a 0.25 mm mesh screen in the tailbox,
weighted and sieved at half φ intervals, and then used to calibrate the light table data. To avoid random fluctuations
in sediment transport, we report the bedload transport rate measured by light table at a 5-minute resolution, and
characteristic grain sizes of bedload at 15-minute resolution. A range of methods for the estimation of bed shear stress
has been suggested in the literature (reviewed in Whiting and Dietrich, 1990). In this study, the shear stress is estimated
using the depth-slope product corresponding to normal (steady and uniform) flow. This method is selected because
the focus of this work is on overall (mean) parameters controlling bed evolution; in addition, the water was too shallow
to use an ADV. The water surface slope, rather than bed slope, is implemented in the calculation of shear stress, with
the consideration that water surface slope is closer to the friction slope and also has less random fluctuations than bed
slope.

The frequency of measurements during the hydrograph phase is also plotted in Fig. 1(a), with the point gauge

measurements conducted every 30 minutes, the trap weighting/sampling conducted every hour, and the DEM/Wolman
measurements by laser scan/photograph conducted every 2 hours (i.e. at the beginning/end of each stage of the
hydrograph). For each measurement of DEM/Wolman, we stopped the pump instantaneously and let the flow slowly
lower and then stop to allow for the bed to be scanned by a laser and photographed. The time interval between the
stop of the pump and the stop of the flow was about 3 to 4 minutes. To avoid the influence of the following rising
discharge, all subsequent measurements were taken after the flow became stable. The frequency of measurement
during the conditioning phase was adjusted in each experiment in accordance with the duration of the conditioning
phase, and is therefore not plotted in Fig. 1.

The uncertainties associated with the measurement are also studied. For the uncertainties of the standard

deviation of bed elevation, we scanned the floor of the flume twice and calculated the standard deviations of the
scanned DEM. The floor of the flume was horizontal and flat, with no sediment on the bed. Theoretically, the standard
deviation of the DEM should be zero. Therefore, the calculated standard deviations of the flume floor are regarded as
an estimation of the uncertainties of our calculations during experiment. To estimate the uncertainties of the bed
surface GSD, for each measurement the Wolman method was implemented 5 times on the same photograph, with 100
samples/counts each time. The 5 measured GSDs for each time interval were used to calculate the mean and standard
deviation of the bed surface texture (in terms of $D_{s10}$, $D_{s50}$, and $D_{s90}$). To estimate the uncertainties of the light table
method, we compare the data measured by the light table with the data measured by the sediment trap, in terms of
both sediment transport rate and the characteristic grain sizes of sediment load. To estimate the variations of the
measured/calculated data, we calculate their coefficient of variation (cv), defined as the ratio of the standard deviation
to the mean value.
**3 Experimental results**

Table 1 presents an overall schematization of the experimental results, including water surface slope, flow

depth $h$, Froude number $F_r$ ($F_r = u / (gh)^{0.5}$), where $u$ is depth-averaged flow velocity), bedload transport rate $Q_s$, shear
stress $\tau_b$, $D_{50}$ and $D_{90}$ of bed surface ($D_{s50}$ and $D_{s90}$), $D_{50}$ and $D_{90}$ of bedload ($D_{l50}$ and $D_{l90}$), and Shields number $\tau^*_{s50}$
for a given $D_{s50}$. Here $D_{90}$ denotes the grain size such that 90% is finer, and $D_{50}$ denotes the grain size such that 50%
is finer.
**3.1 Channel adjustment**

In this section, we present the channel adjustments during each experiment. Figure 3 shows the difference of

longitudinal DEM averaged over the cross section, which can represent the adjustment of channel topography during
different periods of the experiment. The DEM averaged over the cross section is used here to study the overall

aggradation/degradation of the channel. For reference, detailed information about the DEM at different times during the experiment is provided in the Supporting Information, with REF6 (15) as an example. From Fig. 3(a) we can see that for each experiment, evident degradation occurs during the first 15 minutes, especially at the upstream end of the flume. This is due to the fact that no sediment supply is implemented during the conditioning period, and also the initial bed material is relatively loose. From 15 minutes until the end of the conditioning phase (as shown in Fig. 3(b)), no evident aggradation/degradation is observed for any experiment, indicating that most of the adjustment of channel topography during the conditioning phase has been accomplished within the first 15 minutes. For Step 1 of the hydrograph (as shown in Fig. 3(c)), no evident aggradation/degradation is observed for any of the experiments (with the mean difference of bed elevation $\Delta z_b$ less than $\pm 1$ mm, as shown in Table 1), except for REF7 (0.25) which has the shortest conditioning phase and experienced a mean degradation of 4.8 mm over the whole bed channel. Similarly, the channel keeps relatively stable during Step 2 of the hydrograph for all experiments (as shown in Fig. 3(d)), with no evident aggradation/degradation being observed (the mean difference of bed elevation $\Delta z_b$ is less than $\pm 1$ mm for all experiments). With the increase of flow discharge, some degradation (with a magnitude of about $10 \sim 20$ mm) can be observed in Step 3 for all experiments at the upstream end of the channel, as shown in Fig. 3(e). Such degradation becomes more evident over the entire channel in Step 4 of the hydrograph, when flow discharge reaches its peak value. This is in agreement with the values of $\Delta z_b$ presented in Table 1.

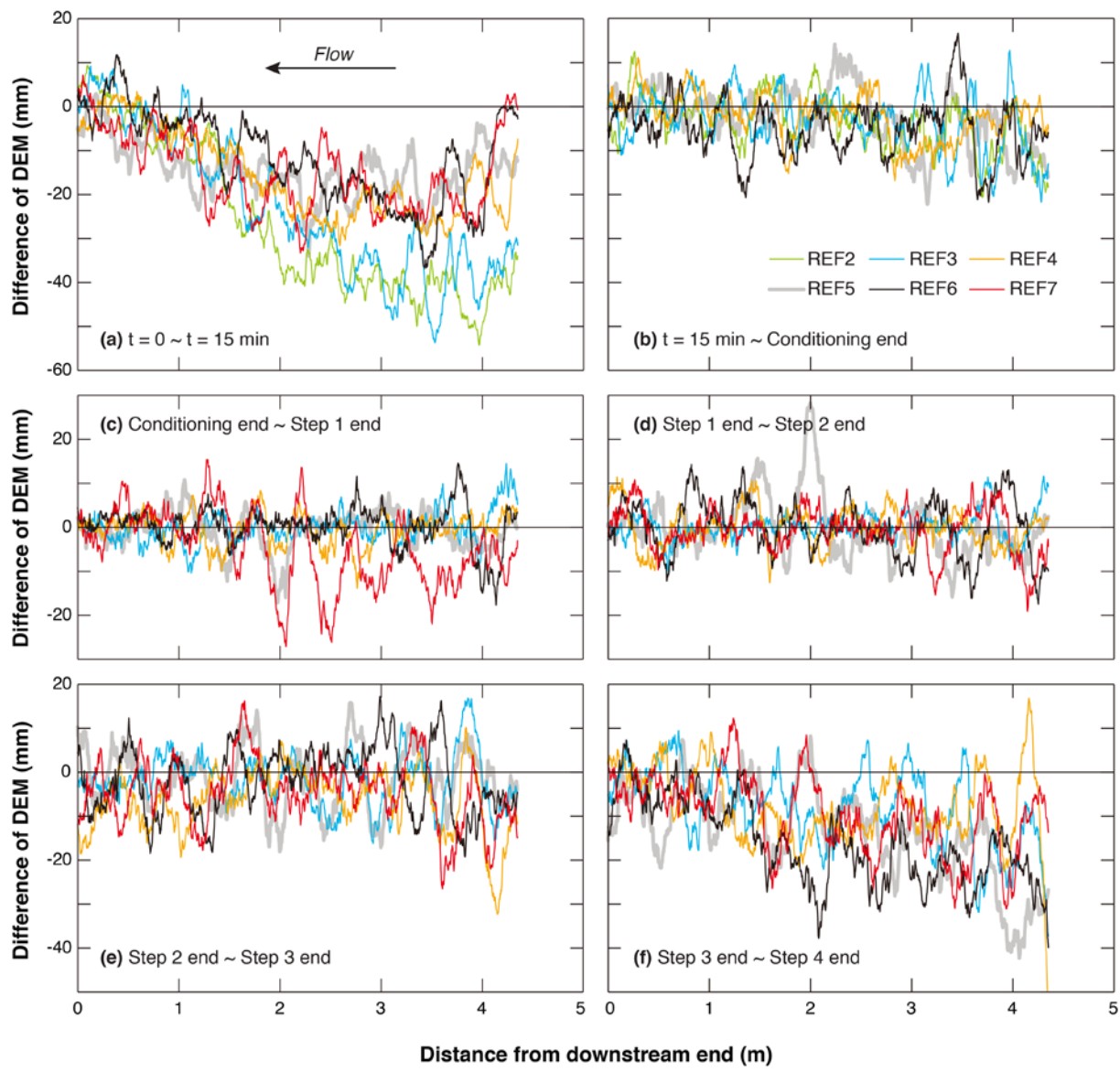

**Figure 3.** Spatial distribution of elevation difference from cross-sectionally averaged longitudinal DEM during the
experiment: (a) from beginning of experiment to $t$ = 15 minutes; (b) from $t$ = 15 minutes to the end of conditioning
phase; (c) from the end of conditioning phase to the end of Step 1 of hydrograph phase; (d) from the end of Step 1 to
the end of Step 2 of the hydrograph phase; (e) from the end of Step 2 to the end of Step 3 of the hydrograph phase; (f)
from the end of Step 3 to the end of Step 4 of the hydrograph phase.
Figure 4 shows the temporal variation of the standard deviation of bed elevation, which is often scaled with
the bed roughness for gravel-bed rivers (see Chen et al. (2020) for a detailed discussion on this topic), over the length
of the erodible bed during the experiment. Results show that the standard deviation of bed elevation is relatively small
at the beginning of the experiments (corresponding to a relatively smooth bed depending on the way we prepared the
initial bed), but increases notably within 15 minutes after the start of the conditioning phase. Such an increase of the
standard deviation of bed elevation is accompanied by significant degradation during the first 15 minutes, as shown
in Fig. 3(a). The standard deviation of bed elevation becomes quite stable during the remaining conditioning phase,
as well as during the hydrograph phase, despite the fact that degradation is evident as the flow approaches its peak
value. For the standard deviation of bed elevation during the conditioning phase, we calculate the coefficient of
variation (cv) for REF2 (15), which has the longest conditioning phase. The result shows a value of 0.038 from $t = 15$
minutes to the end of conditioning flow. For the standard deviation of bed elevation during the hydrograph phase, we
calculate the cv for all experiments; the results show that the values of cv vary between 0.031 and 0.075. Besides, the
value of standard deviation is almost identical for each experiment, indicating the period of conditioning phase exerts
little effect on the standard deviation of bed elevation.

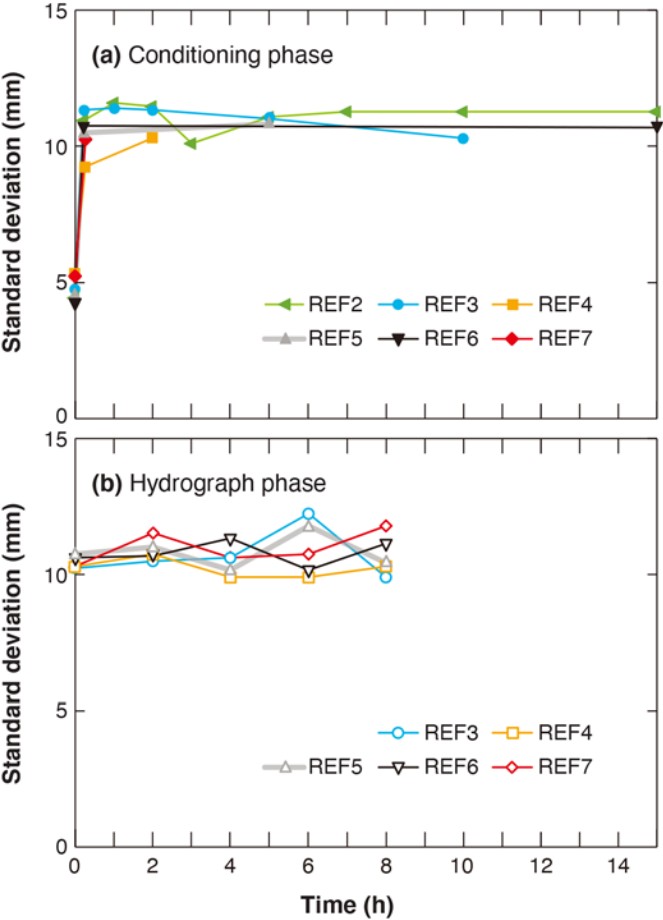

**Figure 4.** Temporal adjustments of standard deviation of bed elevation calculated over the whole erodible bed: (a) the
conditioning phase; (b) the hydrograph phase. The uncertainty of the calculation is in the range of 1.6~2.5 mm, which
is close to the vertical resolution of the laser (1 mm).
Figure 5 shows the temporal variation of the characteristic grain size of bed surface material, as well as an
estimation of the uncertainties associated with measurements of the surface texture. Three parameters are presented
here; $D_{s10}$, $D_{s50}$, and $D_{s90}$. The adjustment of bed surface GSD follows similar trends as the adjustment of standard
deviation of bed elevation. That is, for all experiments the bed surface is fine at the beginning, and experiences a fast
coarsening period during the first 15 minutes (along with the bed degradation in Fig. 3 and the increase of bed

roughness in Fig. 4). The characteristic grain sizes of bed surface remain relatively stable after the first 15 minutes, despite variabilities due to the measurement uncertainty. For REF2 (15) which has the longest conditioning phase, cv (coefficient of variation) values of the mean $D_{s10}$, $D_{s50}$, and $D_{s90}$ (over the five repeated measurements) are 0.15, 0.09, and 0.02 respectively from $t = 15$ minutes to the end of the conditioning flow. It is worth noting that the GSD of bed surface keeps relatively constant even during the hydrograph phase, during which a flood event is introduced in the flume and evident bed degradation is observed. For each experiment, the cv values of the mean $D_{s10}$, $D_{s50}$, and $D_{s90}$ (over the five repeated measurements) are less than 0.13, 0.08, and 0.04 respectively during the hydrograph phase.

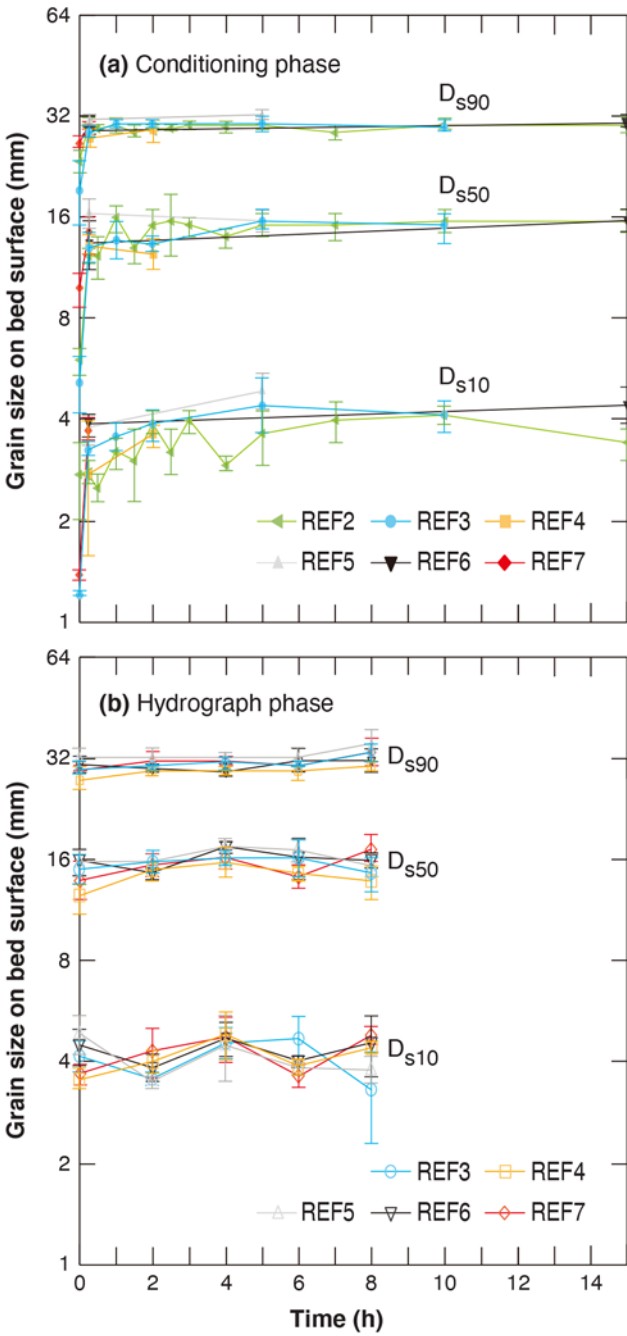

**Figure 5.** Temporal adjustments of characteristic grain sizes of bed surface material calculated over the whole erodible bed: (a) the conditioning phase; (b) the hydrograph phase. Markers show mean values of five repeated Wolman measurements. Range bars show the mean values ± the standard deviations of the five repeated Wolman measurements.

### 3.2 Sediment transport

In Fig. 6 we present the instantaneous sediment transport rate $Q_s$ measured by the light table during each experiment. Sediment transport is reported every 5 minutes, as described in Sect. 2. Accuracy of the results is estimated by comparing the light table data with the data measured by the trap. Results show that for our experiments, the light table method has good accuracy in terms of the sediment transport rate, with an overestimation by 4% on average (111 samples and a standard deviation of 14.5%). 70 out of 111 samples show an accuracy of ±10%, and 93 out of 111 samples show an accuracy of ±20%. Details of this uncertainty analysis are presented in the Supporting Information.

It can be seen in Fig. 6(a) that the temporal variation of sediment transport rate during the conditioning phase follows the same trend in all six experiments. That is, the sediment transport rate decreases significantly during the conditioning phase, with the decreasing rate being very large at the beginning and then gradually dropping. In the first 15 minutes, the sediment transport rates drop from more than 500 kg/h to less than 100 kg/h. Afterwards, it takes about another 2 hours for the sediment transport rates to drop to close to 1 kg/h. The sediment transport rate eventually approaches a small and relatively constant value after about 8 hours of conditioning flow. For REF2 (15) and REF6 (15) which have the longest conditioning phase, the sediment transport rates between $t = 8$ hour and the end of conditioning phase ($t = 15$ hour) show mean values of 0.35 kg/h (standard deviation = 0.22 kg/h) and 0.37 kg/h (standard deviation = 0.24 kg/h), respectively. Nevertheless, there are random high points in the sediment transport rate even after 8 hours, despite no sediment feed from the inlet. These spikes imply that partial destruction (or reorganization) of the bed structure occurs even after a long duration of conditioning.

Previous researchers (Haynes and Pender, 2007; Masteller and Finnegan, 2017) have suggested that an exponential function can be implemented to describe such a decrease of sediment transport rate under conditioning flow. Additional analysis is implemented in the Supporting Information to fit REF2 (15) and REF6 (15) (which have the longest duration of conditioning phase) against a two-parameter exponential function. Results show that the exponential function can describe the general decreasing trend of sediment transport rate during the conditioning phase, except at the beginning of the experiment where the decrease of sediment transport rate is much more significant than that predicted by the exponential function. Readers can refer to the Supporting Information for more details.

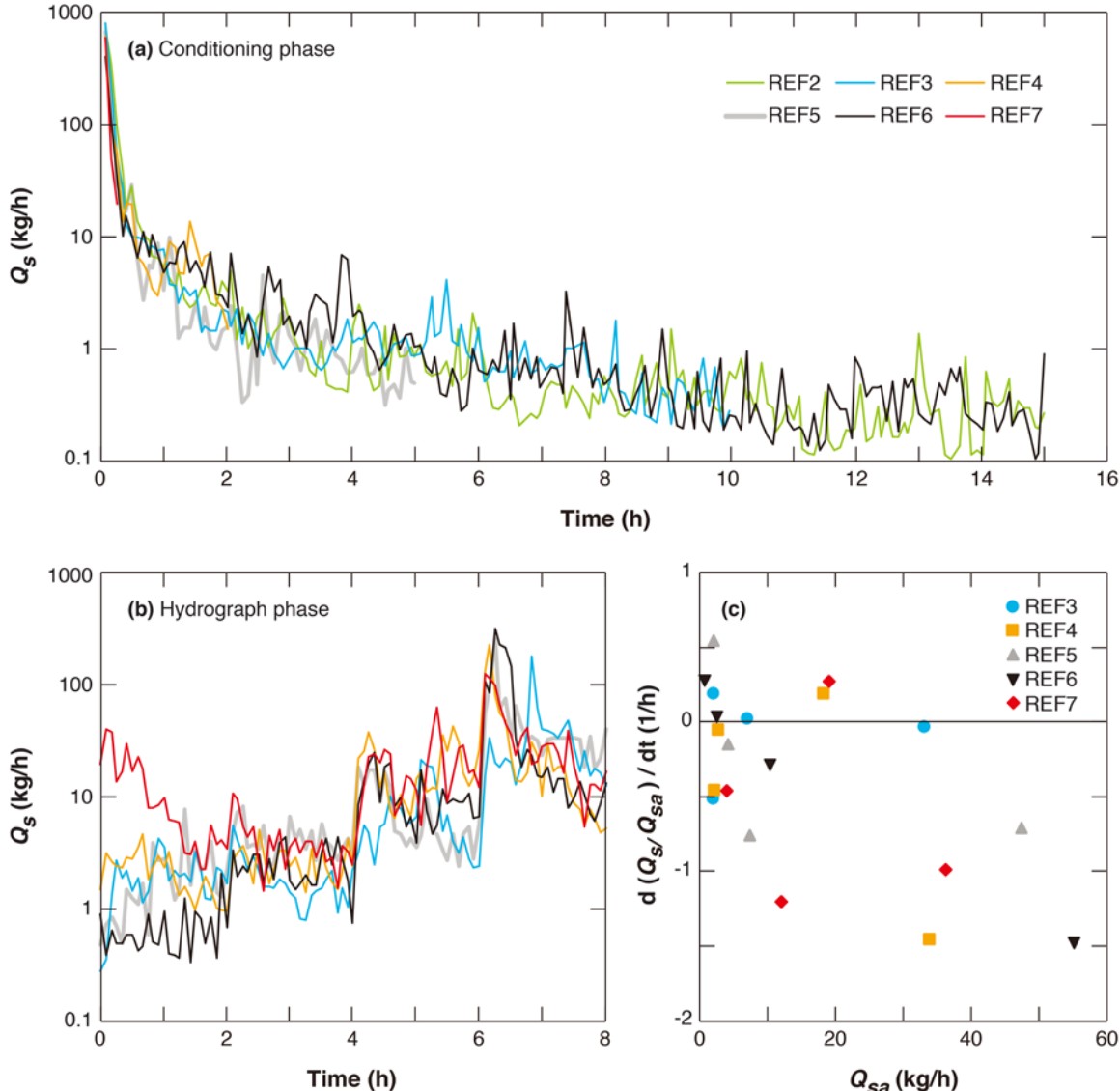

**Figure 6.** Instantaneous sediment transport rate measured by light table during (a) the conditioning phase; and (b) the
hydrograph phase. (c) Intra-step temporal change rate of $Q_s$ normalized against $Q_{sa}$ for each hydrograph step. $Q_s$ is the
sediment transport rate, and $Q_{sa}$ is the averaged sediment transport rate of a given hydrograph step.

Figure 6(b) presents the instantaneous sediment transport rate during the hydrograph phase. Results show
that variation of sediment transport rate among different experiments prevails in the first step of the hydrograph, with
the highest sediment transport rate for the experiment with the shortest conditioning duration (REF7 (0.25)); and the
smallest sediment transport rate for the experiment with the longest conditioning duration (REF6 (15)). Such variation
among experiments, however, diminishes towards the end of Step 1 and is not observed in the following three steps
of the hydrograph, with the line for each experiment collapsing together in the figure. Such adjustments of sediment
transport rate are consistent with the process of channel deformation shown in Fig. 3. That is, for both sediment
transport and channel deformation, results of REF7 (0.25) deviate from other experiments in Step 1 (larger sediment
transport rate and more degradation in REF7 (0.25)), but collapse with other experiments in the following three steps.
Results in Fig. 6(b) also show large variations of sediment transport rate during each step of the hydrograph.
Such intra-step variations of sediment transport rate are investigated in Fig. 6(c), with the $x$ axis being the averaged
sediment transport rate of each step $Q_{sa}$ and the $y$ axis being $d(Q_s/Q_{sa})/dt$. The value of $d(Q_s/Q_{sa})/dt$ is estimated by
linear regression. Here the instantaneous sediment transport rate $Q_s$ is scaled against the average sediment transport
rate of the corresponding step $Q_{sa}$, in order to facilitate the comparison among different hydrograph steps.
Results in Fig. 6(c) shows that a large fraction of the data (11 out of 20) exhibits a decreasing trend in time
for $Q_s$ (i.e. a negative value in vertical coordinate). Basically, the larger the averaged sediment transport rate $Q_{sa}$, the
larger the rate of reduction in $Q_s$. Ferrer-Boix and Hassan (2015) observed similar declines in sediment transport
during their water pulses experiments. They attributed this to (1) the presence of bed structures, which could have
reduced skin friction up to 20% and (2) streamwise changes in the patterns of bed surface sorting. Out of 20 datasets,
5 exhibit some temporally increasing trend in $Q_s$ (though not as evident as the decreasing trend mentioned before).
They are REF5 (5), REF3 (10), REF6 (15) during the first step; and REF7 (0.25), REF4 (2) during the third step. This
shows that for the three experiments with long conditioning duration, $Q_s$ is very low at the end of the conditioning
phase, and the first step of the hydrograph sees a temporally increasing trend in $Q_s$. Whereas for the two experiment
with short conditioning phase, $Q_s$ is still high at the end of the conditioning, so that the sediment transport rate keeps
decreasing during the first step, until in the third step an increasing trend in $Q_s$ is observed, at which the water and
sediment supply become evidently higher. The decreasing/increasing trends of $Q_s$ during steps of the hydrograph
reflect the transient adjustments of the bed to the changed water and sediment supply before equilibrium is achieved.
Sediment collected in the trap/tailbox at the flume outlet allows us to plot the total amount of sediment output
during each step of the hydrograph. Fig. 7(a) shows the total sediment output during the entire hydrograph. It can be
seen that the effect of conditioning duration on the total sediment output during the entire hydrograph phase is not
evident: a longer duration of conditioning flow does not necessarily lead to a smaller (or larger) sediment output. The
largest sediment output occurs in REF7 (0.25), which is 55% larger than the sediment output in REF3 (10) which has
the smallest output, but is about the same as (only 4% larger than) the sediment output in REF6 (15). We further
calculate the correlation coefficient between the total sediment output and the duration of conditioning flow, and
obtain a value of $r = -0.14$, indicating that there is almost no correlation between the two parameters.

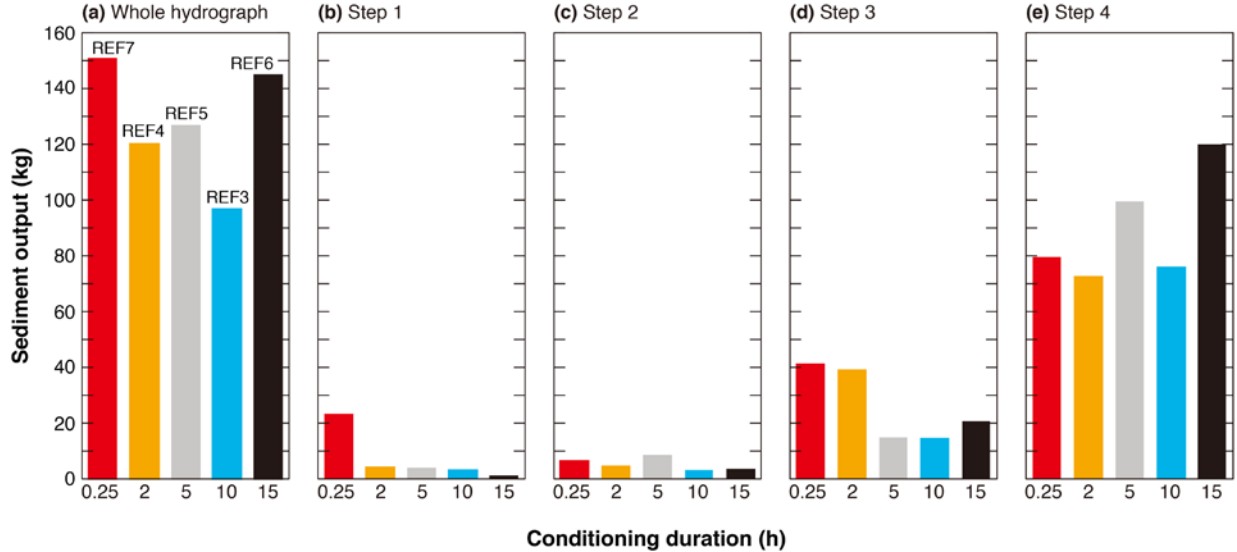

**Figure 7.** Sediment output measured at a trap during (a) the whole hydrograph; (b) Step 1 of the hydrograph; (c) Step
2 of the hydrograph; (d) Step 3 of the hydrograph; (e) Step 4 of the hydrograph.
However, if we study the sediment transport during each step of the hydrograph, we can find that in Step 1
REF7 (0.25) has much larger sediment output than the other experiments, as shown in Fig. 7(b). For Step 1, the
sediment output is 1.1 in REF6 (15), is 3.4~4.4 kg in REF4 (2) REF5 (5) and REF 3(10), and increases sharply to 23.4
kg in REF7 (0.25) (which is more than 20 times of that in REF6 (15)). This agrees with the results for instantaneous
sediment transport rate shown in Fig. 6(b), and shows that the duration of conditioning flow can influence the sediment
transport at the beginning of the subsequent flood, with a longer conditioning phase leading to less sediment transport.
When the duration of conditioning flow is over 2 hours, the subsequent sediment transport rate becomes rather
insensitive to further increase of conditioning duration, indicating that the reorganization of the river bed under
conditioning flow is mostly finished within 2 hours. The effects of stress history on subsequent sediment transport can
hardly be observed during Step 2 of the hydrograph (Fig. 7(c)). Sediment output in REF7 (0.25) reduces significantly
to similar magnitude of other experiments, because most of the loose bed material in REF7 (0.25) has been moved by
the end of Step 1. More specifically, the volumes of sediment output in this step range between 3.1 kg and 8.6 kg,
with the largest output occurring in REF5 (5) and the minimum output occurring in REF3 (10). We further calculate
the correlation coefficient between sediment output and conditioning duration and obtain a value of $r = -0.61$,
indicating that a longer conditioning duration can no longer lead to a larger sediment output in this step. In Step 3 of
the hydrograph (Fig. 7(d)), sediment output in REF7 (0.25) and REF4 (2) is larger than in other 3 experiments which
have longer conditioning phases. But in this step the sediment output in REF7 (0.25) is no more than three times that
of the sediment output in REF3 (10), which has the minimum sediment output. This difference of sediment output
among experiments is not as significant as in Step 1. In the last step of the hydrograph, with the flow discharge and
sediment supply approaching their peaks, the difference in sediment output among the five experiments again becomes
small, with the values ranging between 72.1 kg in REF4 (2) and 119.6 kg in REF6 (15). This demonstrates that little
influence of stress history remains in this step.

Figure 8 shows the temporal variation of the grain size distribution of the bedload. Here $D_{l10}$, $D_{l50}$, and $D_{l90}$

denote grain sizes such that 10%, 50%, and 90% are finer in the bedload, respectively. Accuracy of the measurements
is estimated by comparing the light table data with the trap data. Results show that for our experiments, the light table
method has good accuracy in terms of the median size of bedload ($D_{l50}$), with an overestimation by 3% on average
(111 samples and a standard deviation of 40.1%). Measurements of $D_{l10}$ and $D_{l90}$ show less accuracy, with an
underestimation by 20% on average (111 samples and a standard deviation of 39.0%) for $D_{l10}$ and an overestimation
by 30% on average (111 samples and a standard deviation of 26.5%) for $D_{l90}$. Details concerning this uncertainty
analysis are presented in the Supporting Information.

The value of $D_{l10}$ shows a decreasing trend during the conditioning phase (Fig. 8 (a)), with a value of more

than 2 mm at the beginning to about 0.6 mm after 15 hours, in spite of the large fluctuations before 8 hours. The
decrease of $D_{l10}$ reflects an increase in the fraction of the finest sediment in bedload. In the first two steps of the
hydrograph (Fig. 8(b)), the value of $D_{l10}$ is relatively stable for experiments with long conditioning phases (i.e., REF6
(15) and REF3 (10)), but shows a decreasing trend along with fluctuations for experiments with short conditioning
phases (i.e., REF7 (0.25), REF4 (2), and REF5 (5)). The last two steps of the hydrograph see an evident increase in
the value of $D_{l10}$ compared with the first two steps, due to the increase of flow discharge and sediment supply (Fig.
8(b)). We note that such an increase in the $D_{l10}$ is larger than the standard deviation of measurements, as shown above.

Figures 8(c) and 8(d) show the temporal variation of $D_{l50}$. Compared with that of $D_{l10}$, the temporal variation

of $D_{l50}$ shows more significant fluctuations during the conditioning phase (especially after $t = 10$ hour), as well as at
the beginning of the hydrograph. This can be shown by the coefficient of variation (cv) of the grain size. For the
conditioning phase (after $t = 10$ hour), the cv of $D_{l10}$ show an average value of 0.05 whereas the cv of $D_{l50}$ show an
average value of 1.44. For Step 1 of the hydrograph phase, the cv of $D_{l10}$ show an average value of 0.35 whereas the
cv of $D_{l50}$ show an average value of 0.66. For Step 2 of the hydrograph phase, the cv of $D_{l10}$ show an average value of
0.12 whereas the cv of $D_{l50}$ show an average value of 0.54. As for the temporal variation of $D_{l90}$ (in Figs. 8(e) and
8(f)), the fluctuations are still significant, with the average cv being 0.61, 0.34, 0.27 for the conditioning phase (after
$t = 10$ hour), Step 1 of hydrograph phase, and Step 2 of hydrograph phase, respectively. Besides, there is no significant
increase of decrease of $D_{l90}$ during the experiment. This indicates that the transport of the coarsest sediment is not
sensitive to the variation of our experimental conditions. The more significant fluctuations in $D_{l50}$ and $D_{l90}$ might be
attributed to the fact that during relatively low flow coarse sediment is more likely to be near the threshold of motion
and move intermittently, e.g. as individual grains, as opposed to the more continuous movement for fine sediment.
These fluctuations gradually diminish with the increase of flow and sediment supply, as the static armor on bed surface
transits to mobile armor and the movement of coarse grains become more continuous.

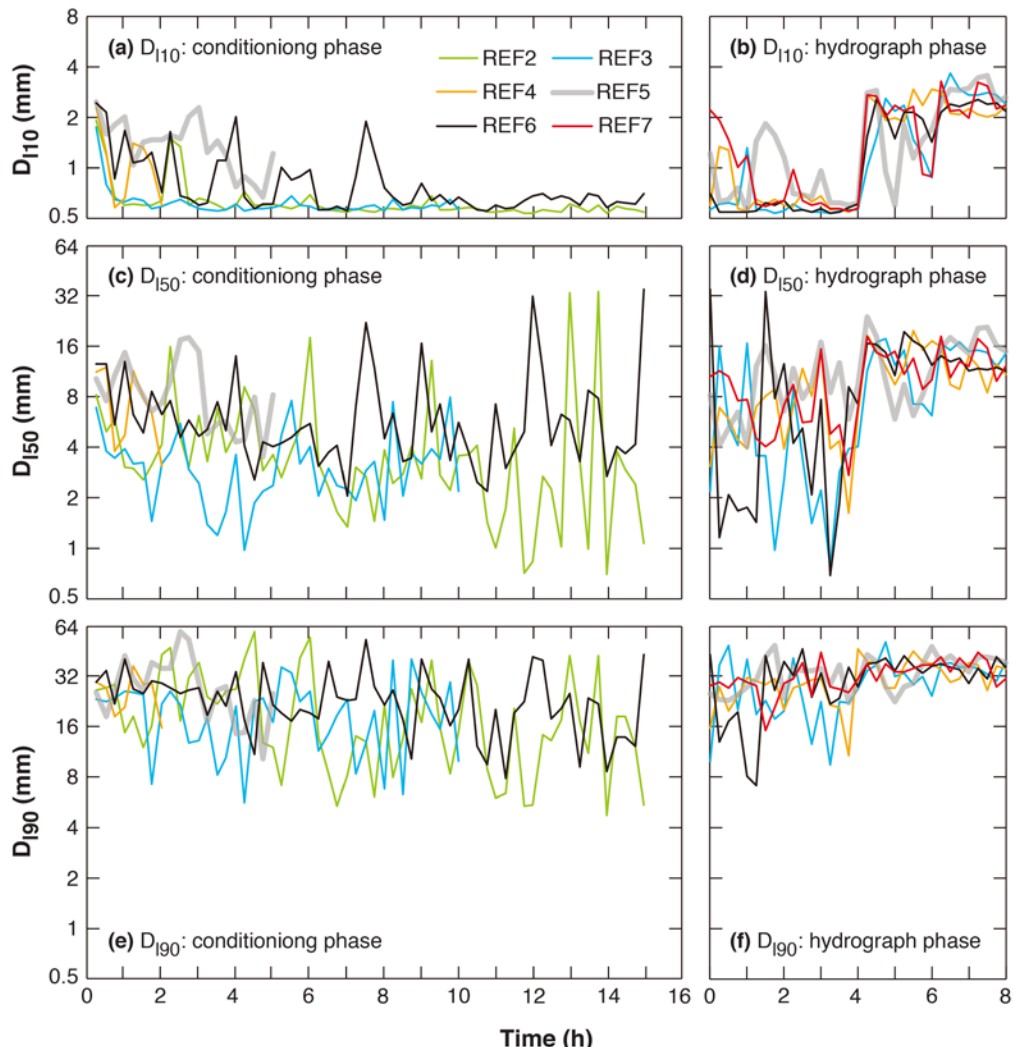

**Figure 8.** Temporal adjustments of characteristic grain sizes of bedload. (a) $D_{l10}$ during conditioning phase; (b) $D_{l10}$
during hydrograph phase; (c) $D_{l50}$ during conditioning phase; (d) $D_{l50}$ during hydrograph phase; (e) $D_{l90}$ during
conditioning phase; (f) $D_{l90}$ during hydrograph phase.
With the fractional sediment transport rate measured by the light table, we also analyze the sediment mobility
of each size range during the experiment. Results show that sediment transport rate is characterized by equal mobility
(i.e., the GSD of sediment load matches the GSD of sediment on bed surface) at the beginning of the conditioning
phase, but moves to partial/selective mobility after a relatively long conditioning phase as well as during the first two
steps of the hydrograph. However, with the increase of flow discharge and sediment supply, the sediment transport
regime gradually returns to equal mobility during the last two steps of the hydrograph. Details of the analysis are
presented in the Supporting Information.

## 4 Discussion

### 4.1 Threshold of sediment motion in experiments

The threshold of sediment motion is a key parameter for the prediction of bedload transport. Previous studies on the stress history effect often start with a conditioning flow that is below the threshold of motion, and then gradually increase the flow discharge, so that the threshold of motion can be directly estimated in the experiment (e.g., Monteith and Pender, 2005; Masteller and Finnegan, 2017; Ockelford et al., 2019; etc.). Because our experiments implement a conditioning flow which can mobilize sediment (sediment transport at the beginning of the conditioning phase is especially large), the threshold of motion cannot be observed directly in the experiment. Here we follow the method applied in Hassan et al. (2020), and estimate the threshold of sediment motion with the Wong and Parker (2006) sediment transport relation, which is a revision of the Meyer-Peter and Müller (1948) relation.

We use the Wong and Parker (2006) relation, which maintains the exponent 1.5, of Meyer-Peter and Muller (1948):

$$q_s^* = 3.97\left(\tau_{s50}^* - \tau_c^*\right)^{1.5} \tag{1}$$

$$q_s^* = \frac{q_s}{\sqrt{RgD_{s50}}\,D_{s50}} \tag{2}$$

$$\tau_{s50}^* = \frac{\tau_b}{\rho g R D_{s50}} \tag{3}$$

$$\tau_b = \rho g h S_w \tag{4}$$

where $q_s^*$ is the dimensionless bedload transport rate (Einstein number) defined by Eq. (2), $\tau_{s50}^*$ is the Shields number for surface median grain size $D_{s50}$ defined by Eq. (3), $\tau_b$ is the flow shear stress calculated using the depth-slope product (Eq. (4)), $\tau_c^*$ is the critical Shields number for the threshold of sediment motion, $q_s$ is the volumetric sediment transport rate per unit width; $h$ is water depth, $S_w$ is water surface slope, $R = 1.65$ is the submerged specific gravity of sediment, $g = 9.81$ m/s$^2$ is the gravitational acceleration and $\rho = 1000$ kg/m$^3$ is the water density. Wong and Parker (2006) proposed a value of 0.0495 for $\tau_c^*$ in Eq. (1). Here we obtain $q_s^*$ and $\tau_{s50}^*$ from the measured data of the experiments, and back calculate the value of $\tau_c^*$ using Eq. (1). It is worth mentioning that in Hassan et al. (2020) three different methods, including the method as described above, are applied to estimate the threshold of sediment motion. Estimation with the three different methods shows very similar temporal trend and variability.

Figure 9(a) shows the values of $q_s^*$ vs. $\tau_{s50}^*$ for each experiment, along with the Wong and Parker (2006) type relation (Eq. (1)) with various values for $\tau_c^*$ (from 0.04 to 0.09). It can be seen from the figure that the measured sediment transport rate is relatively low, with most points below the dimensionless value of 0.001. This indicates that the Shields number in our experiment is slightly larger than the critical Shields number, a state that is typical for gravel-bed rivers (Parker, 1978). The four points with dimensionless transport rate above 0.001 are all at the beginning

of the conditioning flow ($t$ = 15 minutes). The values of $q_s^*$ basically show an increasing trend with the increase of $\tau_{s50}^*$, with the correlation coefficient between $\tau_{s50}^*$ and $\log(q_s^*)$ (consistent with the semi-log scale of Figure 9(a)) being 0.58. Besides, the values of critical Shields number $\tau_c^*$ shown in Figure 9(a) cover a rather wide range (from less than 0.06 to larger than 0.09).

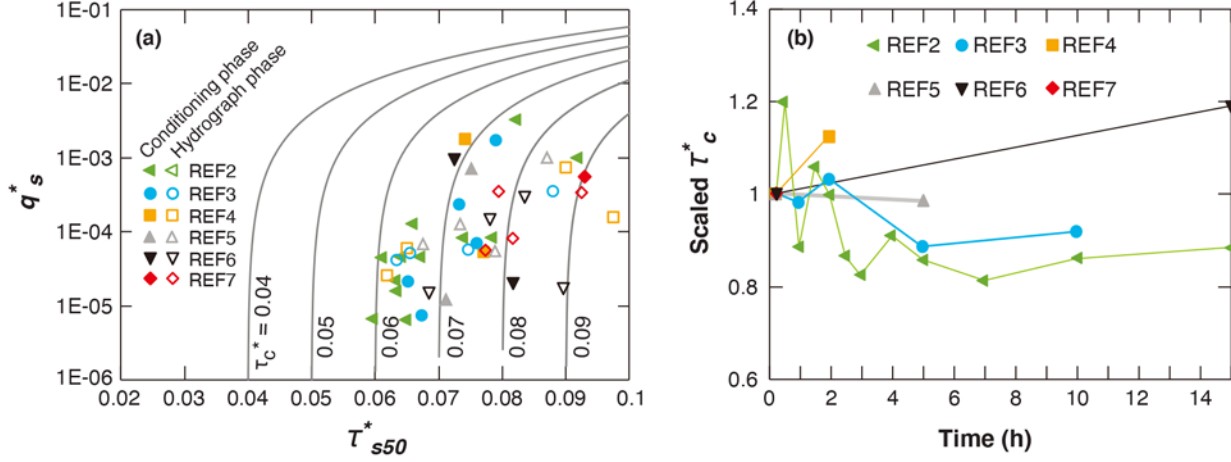

**Figure 9.** (a) Dimensionless sediment transport rate $q_s^*$ vs. Shields number $\tau_{s50}^*$ using surface median grain size for measured transport rates (points). Also shown are lines for the Wong and Parker (2006) type equation (Eq. 1) using different values for $\tau_c^*$. (b) Temporal adjustment of scaled $\tau_c^*$ ($\tau_c^*$ over $\tau_c^*$ at 15 minutes) during the conditioning phase. Here $\tau_c^*$ is back-calculated using Eq. (1) (Wong and Parker (2006) type relation).

Table 2 shows the values of $\tau_c^*$ back-calculated at the beginning ($t$ = 15 minutes) and the end of the conditioning phase in each experiment. The back-calculated values of $\tau_c^*$ vary in the range 0.065~0.090 for the conditioning phase, which is well above the value of 0.0495 as recommended by Wong and Parker (2006). Lamb et al. (2008) demonstrated that critical shear stress can become larger for large bed slope, and they proposed a relation which considers the effect of bed slope,

$$\tau_c^* = 0.15 S_b^{0.25} \tag{5}$$

where $S_b$ is bed slope. For comparison, Table 2 also shows the values of $\tau_c^*$ calculated by Eq. (5). Results shows that for the conditioning phase of our experiments, $\tau_c^*$ calculated by Eq. (5) is above 0.06, which is much higher than the recommended value of Wong and Parker (2006). Besides, the $\tau_c^*$ values predicted by the Lamb et al. (2008) relation show little variability among different experiments, compared with the values back-calculated with equation (1) based on experimental data. More specifically, the cv values are 0.032 at $t$ = 15 minutes and 0.031 at the end of the conditioning phase for $\tau_c^*$ predicted by Lamb et al. (2008) relation, but become 0.10 at $t$ = 15 minutes and 0.12 at the end of the conditioning phase for $\tau_c^*$ back-calculated with equation (1) using measured data. Such discrepancies could be ascribed to the fact the relation of Lamb et al. (2008) considers only the influence of bed slope, without

considering the effects of other mechanisms like organization of surface texture, infiltration of fine particles, etc.
These potential effects are discussed in more detail in Section 4.2.
Here we also estimate the uncertainties associated with the calculation of $\tau_c^*$. For $\tau_c^*$ back-calculated with
equation (1), the global uncertainty is estimated by combining the uncertainties of each parameter involved in the
calculation, i.e. water depth $h$, water surface slope $S_w$, sediment transport rate $q_s$, and surface median grain size $D_{s50}$.
The applied ranges of $h$ and $S_w$ are the measured values plus/minus the errors associated with the gauge point. The
applied ranges of $q_s$ and $D_{s50}$ are the measured values plus/minus the standard deviations as reported in Section 3.
Results of the uncertainties are presented in the brackets in Table 2. For the $\tau_c^*$ values calculated with Equation (5),
the uncertainties are only from the bed slope $S_w$ (which is related with the resolution of point gauge), and is less than
$\pm1\%$ according to our estimates. Therefore, the uncertainty of $\tau_c^*$ calculated with the Equation (5) is not presented in
the table. It can be seen from Table 2 that the values of $\tau_c^*$ calculated with the Equation (5) are mostly within the
uncertainty range of $\tau_c^*$ back-calculated with Eq. (1), with the values closer to the lower bound of the uncertainty
range.

**Table 2.** Values of $\tau_c^*$ at the beginning ($t = 15$ minutes) and the end of conditioning phase in each experiment. Here
$\tau_c^*$ is back-calculated with Eq. (1). Also shown here are values of $\tau_c^*$ estimated with the equation of Lamb et al. (2008)
for comparison. Values in the brackets denote the range of uncertainty associated with the $\tau_c^*$ values back-calculated
with Eq. (1).

| | $t = 15$ minutes | | End of conditioning | |
|---|---|---|---|---|
| | Back-calculated by Eq. (1) | Lamb et al. (2008) | Back-calculated by Eq. (1) | Lamb et al. (2008) |
| REF2 (15) | 0.073 (0.064, 0.083) | 0.063 | 0.065 (0.057, 0.074) | 0.061 |
| REF6 (15) | 0.068 (0.053, 0.089) | 0.066 | 0.081 (0.072, 0.093) | 0.063 |
| REF3 (10) | 0.073 (0.061, 0.088) | 0.061 | 0.067 (0.058, 0.079) | 0.060 |
| REF5 (5) | 0.072 (0.061, 0.085) | 0.065 | 0.071 (0.062, 0.081) | 0.063 |
| REF4 (2) | 0.068 (0.059, 0.079) | 0.061 | 0.077 (0.066, 0.090) | 0.062 |
| REF7 (0.25) | 0.090 (0.075, 0.109) | 0.066 | 0.090 (0.075, 0.109) | 0.066 |


In Fig. 9(b), we plot the scaled $\tau_c^*$ during the conditioning phase of our experiments. For each experiment,
the scaled $\tau_c^*$ is calculated as the ratio between $\tau_c^*$ and the corresponding $\tau_c^*$ at $t = 15$ minutes. $\tau_c^*$ implemented here
is back-calculated with Eq. (1). The scaled $\tau_c^*$ collapses on a value of unity at $t = 15$ minutes (i.e., the first point of
each experiment). It can be seen from the figure that different trends are exhibited for the adjustment of $\tau_c^*$ from $t =$
15 minutes to the end of conditioning phase, with REF2 (15) and REF3 (10) exhibiting a decreasing trend, REF5 (5)
exhibiting very slight changes, and REF4 (2) and REF6 (15) exhibiting an increasing trend. The decrease of $\tau_c^*$ in
REF2 (15) an REF3 (10) is accompanied by a reduction of Shields number $\tau_{s50}{}^*$, mainly due to the increase of surface
median grain size $D_{s50}$. Moreover, the variation of back-calculated $\tau_c{}^*$ is mostly within a range of ±20%, in agreement
with our observation that variation of bed topography and bed surface texture become insignificant after 15 minutes.
It should be noted that $\tau_c{}^*$ cannot be back-calculated using Eq. (1) within the first 15 minutes of the conditioning phase,
since the information for flow depth, water surface slope and bed surface GSD is not available. Nevertheless, we
expect the adjustment of $\tau_c{}^*$ could be evident within the first 15 minutes, since the adjustments of both bed topography
and bed surface are significant during this period (as shown in Sect. 3.1).
**4.2 Implications and limitations**

Previous research has shown that antecedent conditioning flow can lead to an increased critical shear stress

and reduced sediment transport rate during subsequent flood event (Hassan and Church, 2000; Haynes and Pender,
2007; Ockelford and Haynes, 2013; Masteller and Finnegan, 2017). Our flume experiments also show a reduction in
sediment transport rate, especially at the beginning of the hydrograph, in response to the implementation of antecedent
conditioning flow (as shown in Fig. 6(b) and Fig. 7). However, our results are different from previous research in that
the influence of antecedent conditioning flow is found to last for a relatively short time at the beginning of the
following hydrograph, and then gradually diminish with the increase of flow intensity as well as sediment supply (Figs.
6 and 7). Such results indicate that increasing flow intensity and sediment supply during a flood event can lead to the
loss of memory of stress history. A similar phenomenon was observed by Mao (2018) in his experiment, where
sediment transport during a high-magnitude flood event was not much affected by the occurrence of lower-magnitude
flood event before. Besides, the subsequent hydrograph leads to evident bed degradation (Fig. 3) and increase of
sediment transport rate (Figs. 6 and 7), but does not lead to evident change of surface texture or break of the armor
layer (Fig. 5). This is in agreement with the observation of Ferrer-Boix and Hassan (2015) during experiments of
successive water pulses.

Our results have practical implications for mountain gravel bed rivers. The importance of conditioning flow

has long been discussed in the literature, and researchers have suggested that the stress history effect be considered in
the modeling and analysis of gravel bed rivers. For example, previous research states that existing sediment transport
theory for gravel bed rivers (e.g., Meyer-Peter and Müller, 1948; Wilcock and Crowe, 2003; Wong and Parker, 2006;
etc.) might lead to unrealistic predictions if the stress history effect is not taken into account (Masteller and Finnegan,
2017; Mao, 2018; Ockelford et al., 2019). Our results indicate that the stress history effect is important and needs to
be considered for low flow as well as the beginning of the flood event, but becomes insignificant as the flow gradually
approaches high flow discharge.

To explain the effect of stress history, Ockelford and Haynes (2013) has summarized the following possible

mechanisms. (1) Vertical settling during the conditioning flow consolidates the bed into a tighter packing arrangement
which is more resistant to entrainment. (2) Local reorientation and rearrangement of surface particles provide a greater
degree of imbrication, less resistance to fluid flow, as well as direct sheltering on the bed surface. (3) The infiltration
of fines into low-relief pore spaces can further increase the bed compaction. In the experiment of Masteller and
Finnegan (2017), it was found that the most drastic changes during conditioning flow are manifest in the extreme tail
of the elevation distribution (i.e., the reorientation of the highest protruding grains into nearby available pockets) and
go therefore undetected in most bulk measurements (e.g. the mean bed elevation, standard deviation of bed topography,
or the bed surface GSD). They demonstrated that such reorganization of the highest protruding grains can indeed lead
to noticeable differences in the threshold of sediment transport (Masteller and Finnegan, 2017). This might explain
the observation in our experiment that after the first 15 minutes of the conditioning phase, adjustments of the bed
topography and the bed surface GSD become insignificant, but the sediment transport rate as well as its GSD keeps
adjusting consistently.

In our experiments as well as previous experiments that study the effect conditioning flow (e.g., Monteith

and Pender, 2005; Masteller and Finnegan, 2017; Ockelford et al., 2019), no sediment supply is implemented during
the conditioning flow, and the flow can reorganize the bed surface to a state that is more resistant to sediment
entrainment. Therefore, it is straightforward to expect that the conclusions based on our flume experiments to apply
for natural rivers where sediment supply is relatively low during low flow conditions. However, some gravel-bed
rivers have quite active hillslopes, and sediment input from hillslopes to river channel can occur regularly (Turowski
et al., 2011; Reid et al., 2019). Since the sediment material from hillslopes is typically loose and easy to transport,
under such circumstances a long inter-event duration (i.e., low-flow duration) might lead to an enhanced sediment
transport rate in the subsequent flood (Turowski et al., 2011).

It should also be noted that in previous experiment on the stress history effect, conditioning flow is often set

below the threshold of sediment motion. One exception is the experiment of Haynes and Pender (2007) in which the
conditioning flow was above the threshold of motion for $D_{50}$. By implementing conditioning flow with various
durations and magnitudes, they demonstrated that a longer duration of conditioning flow will increase the bed stability
whereas a higher magnitude of conditioning flow will reduce the bed stability. However, since the subsequent flow
they implement to test the bed stability was constant through time, their results did not show how a subsequent flow
event with increasing intensity would affect the stress history. Here we implement a conditioning flow which can
mobilize sediment, especially at the beginning of the conditioning phase during which evident sediment transport
occurs. Moreover, by implementing a subsequent (rising limb of) hydrograph, we find that the stress history can persist
during the beginning of the hydrograph but is eventually erased out as the flow intensity increases. In our experiments,
we varied the duration of conditioning flow by fixing the conditioning flow magnitude. In this sense, how the stress
history formed under various magnitudes of conditioning flow (both above-and below-threshold) would be affected
by a subsequent hydrograph still merits future research.

Recently, Church et al. (2020) drew attention to the reproducibility of results in geomorphology. They

distinguished three levels of "reproducibility", including "repetition", "replication", and "reproduction". In this paper,
the repetition of experimental results is tested by repeating the conditioning phase with the longest duration (REF6
(15) and REF2 (15)). The two experiments show similar results during the conditioning phase in terms of standard
deviation of bed elevation, GSD of bed surface, sediment transport rate, and GSD of sediment load. However, the
reproduction of the experimental results, which requires independent tests undertaken using different materials and/or
different conditions of measurement, and which is more significant, according to Church et al. (2020), for advancing
of the science, has not been tested in this paper. In this regard, more efforts are needed in future study to test the
reproducibility of the conclusions given in this paper.
**5 Conclusions**
In this paper, the effect of antecedent conditioning flow (i.e., the effect of stress history) on the
morphodynamics of gravel-bed rivers during subsequent floods is studied via flume experimentation. The experiment
described here is designed based on the conditions of East Creek, Canada. The experiment consisted of two phases: a
conditioning phase with constant water discharge and no sediment supply, followed by a hydrograph phase with
hydrograph and sedimentograph. Five runs (REF 3~7) were conducted with identical experimental conditions except
different durations of conditioning phase. Another run (REF 2), which consisted of only the conditioning phase, is
conducted in order to test the reproducibility of experimental results during the conditioning flow. Experimental results
show the following.
●    Adjustments of channel morphology (including channel bed longitudinal profile, standard deviation of bed

elevation, characteristic grain sizes of bed surface material) are evident during the first 15 minutes of the

conditioning phase, but become insignificant during the remainder of the conditioning phase.

●    The implementation of conditioning flow can indeed lead to a reduction in sediment transport during the

subsequent hydrograph, which agrees with previous research.

●    However, the effect of stress history on sediment transport rate is limited to a relatively short time at the beginning

of the hydrograph, and gradually diminishes with the increase of flow discharge and sediment supply, indicating

a loss of memory of stress history under high flow discharge. Also, the effect of stress history on the GSD of

both bed surface and bedload is not evident.

●    The threshold of sediment motion is estimated with the form of the Wong and Parker (2006) relation. The

estimated critical Shields number varies in the range 0.066~0.086 during the conditioning phase (excluding the

first 15 minutes), and is higher than the value recommended by Wong and Parker (2006).

Our study has implications in regard to a wide range of issues for mountain gravel-bed rivers, including
sediment budget analysis, river morphodynamic modeling, water and sediment regulation, flood management, and
ecological restoration schemes.
**Notation**
$D_{l50}$: grain size such that 50 percent in sediment load is finer (similarly $D_{l10}$ is such that 10 percent in sediment load
is finer and $D_{l90}$ is such that 90 percent in sediment load is finer).
$D_{s50}$: grain size such that 50 percent on bed surface is finer (similarly $D_{s10}$ is such that 10 percent on bed surface is
finer and $D_{s90}$ is such that 90 percent on bed surface is finer).
$F_r$: Froude number.
$g$: gravitational acceleration.
$h$: water depth.

569 $Q_s$: sediment transport rate.

570 $q_s$: volumetric sediment transport rate per unit width.

571 $q_s^*$: the dimensionless bedload transport rate (Einstein number).

572 $R$: submerged specific gravity of sediment.

573 $S_b$: bed slope.

574 $S_w$: water surface slope.

575 $\rho$: water density.

576 $\Delta z_b$: mean difference of bed elevation;

577 $\tau_b$: bed shear stress.

578 $\tau_c^*$: critical Shields number for the threshold of sediment motion.

579 $\tau_{s50}^*$: dimensionless shear stress (Shields number) of the $D_{s50}$.

**Data availability**

581 Data used for the analysis can be found at doi: 10.6084/m9.figshare.12758414 (An, 2020).

**Author contribution**

583 Marwan A. Hassan and Xudong Fu designed the research. Carles Ferrer-Boix performed the experiments. Chenge An
584 processed and analyzed the experimental data. Chenge An prepared the manuscript with contributions from all
585 coauthors.

**Competing interests**

587 The authors declare that they have no conflict of interest.

**Acknowledgments**

589   Gary Parker provided constructive comments and helped edit this paper. Maria A. Elgueta-Astaburuaga
590 helped conduct the experiments. Rick Ketler provided support in equipment and data collections. Eric Leinberger
591 provided support in designing the figures. We thank Jens Turowski and another anonymous reviewer for their
592 constructive comments, which helped us greatly improve the paper. This study was funded by the National Natural
593 Science Foundation of China (grants 52009063, U20A20319, 52079095, 41941019, 91747207) and the China
594 Postdoctoral Science Foundation (grant 2018M641368).

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
