# Peer review of "Effect of stress history on sediment transport and channel adjustment"

_Earth Surface Dynamics, 2020_

## Referee Comment (RC1) · Jens Turowski (Referee) · 14 Sep 2020

In the paper, the authors describe experiments to investigate the effects of conditioning flows on sediment transport. They argue that conditioning flows can indeed reduce transport rates, as has been suggested before, but only for a limited time. Memory is lost as sediment supply rates and discharge increase. The paper presents some interesting data, and the topic is timely and is generally suitable for publication in ESurf. There are, however, a few short-comings that should be addressed.

First, the interpretation of the data frequently rests on subjective judgements or undisclosed criteria and arguments. For example, the authors identify trends or 'significant' differences without explaining their criteria or providing suitable statistical tests. Further, errors and uncertainties are not reported or discussed. Given the often large fluctuations, the trends, effects and differences claimed by the reviewers are mostly hard to spot, or they cannot be judged against the uncertainties in the measurements. This makes it hard for the reader to fully understand and believe the conclusions. I ask the authors to provide uncertainty estimates for the key parameters, and to re-assess their interpretation in light of these uncertainties, best with suitable statistical tests.

Second, the reader is not guided through the entire argument and often, the punchlines are not explicitly delivered. For example, in the discussion section 4.1, the authors finish their back-calculation of the threshold of motion with the statement "...indicating that only the slope effect cannot explain the observed range of tau_c* (line 354). The obvious question to ask is: if it is not slope, what is it? I think the authors are trying to address this question in the following paragraph (starting on line 360), but this paragraph contains merely some further statements about the data. As a reader, I am unsure what features of the data I should particularly aware of, how they are interpreted and how this leads to the conclusion of the authors. There are similar problems in other parts of the manuscript. For example, the authors claim that they observe reduced transport rates after the conditioning phase (e.g.,. line 376: "Our flume experiments also show a reduced sediment transport rate in response to the implementation of conditioning flow."), which presumably rests on data shown in Fig. 6c and Fig. 7. These data show large scatter and behavior contrary to the expectation. They and their interpretation are not picked up in the discussion other than as a stepping point to the implications. Statistical tests of presumed similarities and differences are missing, as are error bars. To address these issues, I ask the authors to develop a clear logic with explicitly stated punchlines. In the best case, they can formulate an expectation (e.g., condition leads to an increase in the threshold of motion), a null hypothesis (e.g., threshold is constant), and a suitable statistical test to discriminate them. Then, they can walk the reader through the various observations until the conclusion is reached.

to me it makes sense to differentiate the terminology somewhat. The terminology chosen in the cited papers seems to be motivated by the available data and the chosen approach. For example, Mao looked at the effect of flood events, while Masteller et al. focused more on continuous discharge data. Further, a change in flow does not directly translate into a change in stress, there are non-linearities and feedbacks involved (e.g., non-steady flow, re-organization of the bed). It does make sense to use a stress approach (and therefore terminology) for experiments, but stress information is much less reliable for field measurements.

Maybe the effect of sand supply on gravel transport should be mentioned somewhere in this paragraph; see for example Curran and Wilcock, JHE 2005 (DOI: 10.1061/(ASCE)0733-9429(2005)131:11(961)

also Lenzi, (Step-pool evolution in the Rio Cordon, Northeastern Italy. Earth Surface Processes and Landforms 26: 991–1008, 2001), and Turowski et al., 2009 (DOI: 10.1002/esp.1855).

To investigate the study objectives. . .

. . .not fed. . .

107-108 The approach and reasoning here needs some more detail. What were the simulations? What kind of trial experiments? How was the final feed rate chosen?

136-141 Unclear, lacks detail.

How were images merged? How did you deal with image distortion due to merging or lense distortion effects?

perform

I don't understand how particle sizes were measured.

How were changes quantified?

Please add some information on accuracy and precision of this method.

What does "slowly" mean in this context? Please also explain how an effect of the rising discharge after measurements was avoided.

why the average over the cross section, instead of, for example, the thalweg? Maybe this should be described and justified in the method section.

How were trends assessed here?

How exactly was this assessed? What features did you look for to identify a bedform?

Here, the authors implicitly identify the standard deviation of the bed with the bed roughness. Bed roughness is a technical term in fluvial hydraulics, and although there is evidence that the standard deviation scales with roughness, the terms are not directly equivalent, as they are used here.

what does "almost constant" mean in this context? Can you make this statement quantitative?

please add in symbols to mark the time of the actual measurements.

is it possible to add error bars to these plots?

What does "relatively stable" mean in this context? How was stability assessed?

What does "relatively constant" mean in this context? How was constancy assessed?

Interpretation, to discussion.

Please mark the time of the measurements on the plots and add error bars.

Fig. 6c shows presumably not a derivative, but a ratio of discrete changes. Please change the notation accordingly (for example, by using delta symbols). Please add error bars.

During each of the hydrograph steps, there seems to be decline of transport rates over time. This may be due to a transient adjustment of the bed to the changed hydraulic conditions. It does not seem to me here that an equilibrium is achieved. Can the different stages then be meaningfully compared? How would a transient adjustment affect the interpretations?

...of the sediment transport rate...

How did you assess whether it agrees? What does agreement mean in this context?

Please explain how you detected trends and give corresponding statistics. The interpretation is a little difficult, since there are no error bars for the data.

How did you arrive at this assessment? REF7 is around 50% larger than REF3! It would make sense to add error bars to the measurements and a statistical test to actually show that there is no difference.

please add uncertainties.

with a longer conditioning phase leading...

How did you arrive at this interpretation? Maybe this is just an effect of the scale of the plot? It would be good to add error bars to the data here and some suitable statistical test.

290-298 how did you establish significance and what does 'more significant' mean in this context / how was this assessed?

Please give some indications of the uncertainties of these measurements.

303-308 this needs more detail if it is relevant for the central message of the paper. If not, consider deleting it.

The use of the term 'equal mobility' has become ambiguous. Originally, it meant that the grain size distribution of the transport material matches the grain size distribution of the material found on the bed. However, it is now often used to mean an equal threshold of motion for all grain sizes. It is unclear from the context here which meaning is intended.

transport rate please add error estimates to these calculations.

predictions

This would imply that the stronger trends should be seen in d90 rather than d50, right? How does this expectation compare to the data?

either 'e.g.' or 'etc.', not both

. . .the conditioning flow was. . .

In the present paper. . .

consisted consisted

---

## Referee Comment (RC2) · Anonymous Referee #2 · 7 Oct 2020

Review of 'Effect of stress history on sediment transport and channel adjustment in graded gravel-bed rivers' An. C., Hassan, H., Ferrer-Boix, C and Fu, X. Manuscript Number: esurf-2020-67

This paper presents a series of flume experiments detailing the effect of conditioning flow duration on the sediment flux experienced during a subsequent flood wave. The paper in itself is interesting and would be of interest to the readers of ESurf but at the moment the language is not tight enough and the reader is left to try and fathom out the main take home messaged from each set of data analysis. For example, throughout the results there are phrases like 'increases notably', 'significant degradation' etc but it is not clear whether or not the authors have undertaken statistical analysis to support their results. If they have then the outputs of those statistical analysis need to be

reported. It is also unclear how reproducible the results are – again this needs to be reported. Therefore, this leaves the reader wondering how important the reported trends are, especially given the data is relatively noisy. I would also argue that the majority of the data analysis is relatively basic, which in itself is not a problem, but when coupled with the subjective language used, as discussed above, the main 'story' of the manuscript is lost. That said there are places where I think the analysis could be taken a lot further e.g. analysis of bed surface topography is not really mentioned despite DEMs being collected and mentioned in the methodology. Finally, I have a significant issue with section 4.1 and the back calculation of tau*c using MPP and Wong and Parker since this regression is designed for a small range of known tau*c. Irrespective of this I am not sure of the worth of scaling tau*c from that derived right at the beginning of the experiment – the bed state there is not representative of a true fluvial system, rather is representative of artificial conditions caused by screeding and high sediment transport rates caused by initial scour. So basing analysis off this seems odd and slightly misleading. In this section the authors also say that 'only the slope effect cannot explain the observed range of tau*c' – it is not clear what they mean and given this paper is about steep slope environments this needs careful clarification and expansion.

I really like it when authors include an implications section in their papers and so it is good to see this included in the presented paper however some of the text in this section feels much more like discussion and framing of the authors results within the wider literature of which there was relatively little of in the paper up until that point. So, I think the discussion and implications sections of the paper need re-framing slightly so the discussion section properly frames the results within the wider literature and the implications section talks about the bigger picture and importance of the findings more broadly. If the authors do this I think it will be much clearer to the reader about the new findings which this paper had generated – to make this even easier for the reader there are places where paragraphs could be re-structured such as to lead with the findings from this paper before framing within the previous literature. This will help make it

crystal clear where the additional knowledge is. For these reasons I would suggest a rejection of the paper with a strong encouragement to resubmit once the issues have been addressed. Line by line changes are also suggested below to help the authors revise their manuscript.

Suggested Changes

Line 24 – arguably not just mountain streams

Line 26 – I would consider adding Masteller 2019 JGR paper in here as it seems relevant

Line 28 – what do you mean by average flow regime?

Line 44 – remove etc in the citations

Line 52 – should be Haynes

Line 82 – from reading your methodology I would argue that you don't run experiments which consist of ' extended cycles' – for me that reads as you cycled hydrographs but this is not what you did – instead you ran changed the length of a period of conditioning flow before exposing that bed to the rising limb of the hydrograph

Line 88 – can you be more explicit with what you mean by guided by?

Line 89 – to investigate the study objectives…...

Line 104-106 – so was the flow rate directly scaled?

Line 105 – I don't think that you ran a hydrograph – you ran a rising limb of a hydrograph but not a full hydrograph. The results you would have come up with by running a full hydrograph would have been very different to those which you report here

Line 108- 111 – more justification and reasoning is needed in this section – what are the details of the trail experiments? How exactly were the feed rates chosen?

Line 117 – why was this sediment scaling chosen?

Line 123 – what impact does feeding the bulk sediment rather than feeding a GSD which is representative of the transported sediment have on the surface development?

Line 127 – 129 – how often did you measure the bed surface profile (It is difficult to tell from fig 1) – was this just down the channel centre line?

Line 131- 136 and Line 156 - What impact did draining the flume have on the transport dynamics – did you assess this? How were the DEMs detrended? You need a sentence or more in here justifying why you are using std of elevations as a measure of surface topography

Line 148 – was the shear stress corrected for side walls? If so how?

Line 166 – I would have thought it would have been useful to present the statistical moment analysis of the bed scans so you can properly link the development of the bed surface with some of the sediment dynamics. I think it would also be useful to plot the surface DEM evolution to allow readers to better appreciate and understand how the surface evolves.

Line 168 – I am not sure exactly what you mean by longitudinal DEM – why average it over the cross section? A lot of the previous literature on stress history which has undertaken DEM analysis has shown that the surface develops significant spatial complexity which will be lost by the averaging you have undertaken

Line 180 – what do you mean by further analysis of the DEM? Where is this analysis?

Line 205 – should be noting not noted

Line 205 – what do you mean by keeps relatively flat?

Line 212 – how accurate is the light table?

Line 216 – 217 – again be specific – what do you mean by very large? Gradually dropping? Small and relatively constant?
Line 223 - why is this analysis in the supplemental information? The analysis you have undertaken up to this point in the paper would really seem to benefit from this further analysis

Line 268 and 275 – again be specific and give the statistical outputs if you have undertaken statistical analysis.

Line 279 – I am not sure I agree that the five experiments show similar sediment outputs – there may not be a systematic trend related to condition flow duration but there are certainly differences between them.

Line 303 – 308 – again why is this data in the supplemental information – I would have thought it would have been a really important addition to your paper and provided a lot of useful context from which to hang your discussion

Line 334 – sediment transport rate

Line 337 – what do you mean by basically show an increasing trend – be specific

Line 376 – remove etc from the citations

Line 391 – what implications – can you be specific?

Line 405 – 413 – I am afraid I don't see the relevance of this paragraph to the paper.

Line 415 – more should have been made in the discussion of the comparison between the work presented in the current paper and the results of Haynes and Pender (2007) since this is a very relevant study which would have provided really useful comparators.

Line 419 – again I am not sure I agree with you here – the data presented in this paper has shown that the effects of stress history are effectively cancelled out under higher flows in any subsequent flood. However in line 419 you say 'might be more lasting during subsequent flood' – this seems counter to the rest of the message in the paper.

---

## Author Comment (AC1) · 28 Nov 2020

We appreciate the two reviewers for their constructive comments. Details of our reply are presented in the attachments.

Please also note the supplement to this comment:
https://esurf.copernicus.org/preprints/esurf-2020-67/esurf-2020-67-AC1-supplement.zip

---

## Author Response (AR2)

**Response file to comments**

**Referee #1, Jens Turowski**

1. First, the interpretation of the data frequently rests on subjective judgements or undisclosed criteria and arguments. For example, the authors identify trends or 'significant' differences without explaining their criteria or providing suitable statistical tests. Further, errors and uncertainties are not reported or discussed. Given the often large fluctuations, the trends, effects and differences claimed by the reviewers are mostly hard to spot, or they cannot be judged against the uncertainties in the measurements. This makes it hard for the reader to fully understand and believe the conclusions. I ask the authors to provide uncertainty estimates for the key parameters, and to re-assess their interpretation in light of these uncertainties, best with suitable statistical tests.

We appreciate the reviewer for pointing this out! We make efforts to conduct statistical analyses in the revised manuscript, and we think that this helps improve our paper considerably. The methodology that we implement is described below.

An analysis of the uncertainty of bed elevation standard deviation is now conducted by analyzing the DEMs of the flume floor. The uncertainty of bed surface GSD is now analyzed by implementing the Wolman method 5 times for each measurement. The uncertainty of the light table measurement is analyzed by comparing the light table results with the sediment trap results.

The variations in the data are analyzed with the coefficient of variation (cv, defined as the standard deviation over the mean value). The correlation coefficient $r$ is calculated to study the relation between two parameters (e.g. sediment transport rate and duration of conditioning flow).

The methods we used are explained in detail in the text. See Lines 186-197 of the manuscript with Track Changes.

2. Second, the reader is not guided through the entire argument and often, the punchlines are not explicitly delivered. For example, in the discussion section 4.1, the authors finish their back-calculation of the threshold of motion with the statement "…indicating that only the slope effect cannot explain the observed range of tau_c* (line 354). The obvious question to ask is: if it is not slope, what is it? I think the authors are trying to address this question in the following paragraph (starting on line 360), but this paragraph contains merely some further statements about the data. As a reader, I am unsure what features of the data I should particularly aware of, how they are interpreted and how this leads to the conclusion of the authors. There are similar problems in other parts of the manuscript. For example, the authors claim that they observe reduced transport rates after the conditioning phase (e.g.,. line 376: "Our flume experiments also show a reduced sediment transport rate in response to the implementation of conditioning flow."), which presumably rests on data shown in Fig. 6c and Fig. 7. These data show large scatter and behavior contrary to the expectation. They and their interpretation are not picked up in the discussion other than as a stepping point to the implications. Statistical tests of presumed similarities and differences are missing, as are error bars. To address these issues, I ask the authors to develop a clear logic with explicitly stated punchlines. In the best case, they can formulate an expectation (e.g., condition leads to an increase in the threshold of motion), a null hypothesis (e.g., threshold is constant), and a suitable statistical

test to discriminate them. Then, they can walk the reader through the various observations until the conclusion is reached.

For Line 354, we calculate the cv (coefficient of variation) to compare the variability of the $\tau_c^*$ obtained by two different methods. The relation of Lamb et al. (2008) considers the effect of bed slope on the threshold of motion. Other effects that could influence the threshold of motion were previously discussed in more detail in Section 4.2. We rephrase the text as follows so that the reader can understand more easily. "Besides, the $\tau_c^*$ values predicted by the Lamb et al. (2008) relation show little variability among different experiments, compared with the values back-calculated with equation (1) based on experimental data. More specifically, the cv values are 0.032 at t = 15 minutes and 0.031 at the end of the conditioning phase for $\tau_c^*$ predicted by Lamb et al. (2008) relation, but become 0.10 at t = 15 minutes and 0.12 at the end of the conditioning phase for $\tau_c^*$ back-calculated with equation (1) using measured data. Such discrepancies could be ascribed to the fact the relation of Lamb et al. (2008) considers only the influence of bed slope, without considering the effects of other mechanisms like organization of surface texture, infiltration of fine particles, etc. These potential effects are discussed in more detail in Section 4.2." See Lines 457-464 of the manuscript with Track Changes.

For Line 376, what we meant is that the implementation of a (long) conditioning duration will lead to a reduction of sediment transport at the beginning of the subsequent hydrograph. This effect, however, gradually diminishes with the increase of flow intensity and sediment supply during the hydrograph. This conclusion is supported by Figure 6(b) and Figure 7. We rephrase the text to make this clear. See Lines 502-503 of the manuscript with Track Changes. Quantification analysis is added in Section 3; for example, Lines 343-364 of the manuscript with Track Changes.

As for the issue of uncertainties, please see our reply to Question 1 and other related questions below (both reviewers provided constructive comments on this issue).

3. 44 to me it makes sense to differentiate the terminology somewhat. The terminology chosen in the cited papers seems to be motivated by the available data and the chosen approach. For example, Mao looked at the effect of flood events, while Masteller et al. focused more on continuous discharge data. Further, a change in flow does not directly translate into a change in stress, there are non-linearities and feedbacks involved (e.g., non-steady flow, re-organization of the bed). It does make sense to use a stress approach (and therefore terminology) for experiments, but stress information is much less reliable for field measurements.

The referee's comment is very helpful. We agree with this! We put part of this comment in the paper. See Lines 46-50 of the manuscript with Track Changes.

4. 69 Maybe the effect of sand supply on gravel transport should be mentioned somewhere in this paragraph; see for example Curran and Wilcock, JHE 2005 (DOI: 10.1061/(ASCE)0733-9429(2005)131:11(961).

We add the following sentence in this paragraph. "Besides, the supply of fine sediment (during high discharge events) is also widely observed to enhance the mobilization of coarse sediment (Wilcock et al., 2001; Curran and Wilcock, 2005; Venditti et al., 2010)." See Lines 84-85 of the manuscript with Track Changes.

5. 75 also Lenzi, (Step-pool evolution in the Rio Cordon, Northeastern Italy. Earth Surface Processes and Landforms 26: 991–1008, 2001), and Turowski et al., 2009 (DOI: 10.1002/esp.1855).

We add these two references in the paper. See Lines 79-80 of the manuscript with Track Changes.

6. 89 To investigate the study objectives…

Done.

7. 107 …not fed…

Done.

8. 107-108 The approach and reasoning here needs some more detail. What were the simulations? What kind of trial experiments? How was the final feed rate chosen?

We rephrase the text as "For each step of the hydrograph, the feed rate of sediment was specified to be close to the transport capacity of the flow. Determination of the sediment supply rates was facilitated by a numerical model which was calibrated for similar experimental conditions (Ferrer-Boix and Hassan, 2014)." See Lines 120-122 of the manuscript with Track Changes.

9. 136-141 Unclear, lacks detail.

We add more details in the text. Please see our reply to Questions 10, 12, and 13.

10. 137 How were images merged? How did you deal with image distortion due to merging or lense distortion effects?

We add the following sentence in the text. "To avoid distortion effects due to image merging, the width of the image strips that were stitched to get a composite image was specified as just 2 cm." See Lines 154-155 of the manuscript with Track Changes.

11. 137 perform

Done.

12. 139 I don't understand how particle sizes were measured.

Basically we used the Wolman count method to calculate the grain size distribution. This was done by identifying the grain sizes from the photograph (with a 5 cm grid superimposed on the photograph). We rewrite the text as follows, "The particle size distribution of the bed surface was estimated using the Wolman (point count) method, by identifying the grain size of particles at the

intersections of a 5 cm grid superimposed on the photograph. Individual grains were identified by color. For each experiment, the grain size distribution of the bed surface was calculated at different times to quantify its changes during the experiment." See Lines 155-161 of the manuscript with Track Changes.

**13. 140 How were changes quantified?**

In each run, photographs were taken at different times throughout the experiment to calculate the bed surface grain size distribution. Information about the measurement frequency was given in the paper. See Lines 177-180 of the manuscript with Track Changes. The changes can thus be quantified directly by comparison of the grain size distributions at different times. We rewrite the text as "For each experiment, the grain size distribution of the bed surface was calculated at different times to quantify its changes during the experiment." See Lines 159-161 of the manuscript with Track Changes.

**14. 144 Please add some information on accuracy and precision of this method.**

Zimmerman et al. (2008) have conducted a detailed analysis of the accuracy of this method. They reported that the light table method is accurate and precise for sediment from 2 mm to 45 mm after calibration. Here we also analyze the accuracy of this method for the experiments presented in this paper. What we do is as follows.

To estimate the uncertainties of the light table method, we compare the data measured by the trap and the data measured by the light table, in terms of both sediment transport rate and the characteristic grain sizes of sediment load. This is stated in Section 2 of the main text. See Lines 193-195 of the manuscript with Track Changes.

For the total sediment transport rate, the light table data and the trap data show good agreement. More specifically, the light table overestimates the total sediment transport rates by 4% on average (111 samples and a standard deviation of 14.5%). 70 out of 111 samples show an accuracy of ±10% and 93 out of 111 samples show an accuracy of ±20%. We present the results in both the main text (Lines 271-276 of the manuscript with Track Changes) and the Supporting Information (Section S2).

Uncertainties of the bedload characteristic grain sizes are as follows. The $D_{50}$ of bedload measured by light table show relatively good agreement with that measured by trap. The light table overestimates the $D_{50}$ by 3% on average (111 samples and a standard deviation of 40.1%). The accuracy of the values of $D_{10}$ and $D_{90}$ of the bedload is not as good as that of $D_{50}$. The light table underestimates $D_{10}$ by 20% on average (111 samples and a standard deviation of 39.0%), and overestimates $D_{90}$ by 30% on average (111 samples and a standard deviation of 26.5%). We present the results in both the main text (Lines 366-372 of the manuscript with Track Changes) and the Supporting Information (Section S2).

**15. 156 What does "slowly" mean in this context? Please also explain how an effect of the rising discharge after measurements was avoided.**

We explain this issue in the manuscript. We rephrase the text as "For each measurement of DEM/Wolman, we stopped the pump instantaneously and let the flow slowly lower and then stop

to allow for the bed to be scanned by a laser and photographed. The time interval between the stop of the pump and the stop of the flow was about 3 to 4 minutes. To avoid the influence of the following rising discharge, all subsequent measurements were taken after the flow became stable." See Lines 180-183 of the manuscript with Track Changes.

16. 168 why the average over the cross section, instead of, for example, the thalweg? Maybe this should be described and justified in the method section.

We add the following sentence in the paper. "The DEM over the cross section is used here to study the overall aggradation/degradation of the channel." See Lines 207-208 of the manuscript with Track Changes. For reference, we also add the DEM of bed surface at different time during the experiment. This is presented in the Supporting Information (Section S1)

17. 177 How were trends assessed here?

The "word" trend might be misleading. We rewrite this sentence as "…, with no evident aggradation/degradation being observed". See Line 219 of the manuscript with Track Changes. We also calculate the mean difference of bed elevation to support the conclusion. Results are added in Table 1.

18. 180 How exactly was this assessed? What features did you look for to identify a bedform?

Our determination of the bedform is visually based on the DEM as well as the direct observation of the channel bed. We realized that the statement was not accurate without further quantification. Therefore, we remove the sentence in the text to avoid misunderstanding. See Lines 223-224 of the manuscript with Track Changes. We also suggest the reviewer to refer to the DEM that we add in the Supporting Information (Section S1).

19. 191 Here, the authors implicitly identify the standard deviation of the bed with the bed roughness. Bed roughness is a technical term in fluvial hydraulics, and although there is evidence that the standard deviation scales with roughness, the terms are not directly equivalent, as they are used here.

We change the "bed roughness" to "standard deviation of bed elevation" as the referee pointed out. Our recently published paper on WRR (Chen et al., 2020) supports the idea that standard deviation of bed elevation is a good descriptor for bed roughness of gravel-bed rivers. We rewrite the beginning of this paragraph as follows "Figure 4 shows the temporal variation of the standard deviation of bed elevation, which is often scaled with the bed roughness for gravel-bed rivers (see Chen et al. (2020) for a detailed discussion on this topic), over the length of the erodible bed during the experiment." See Lines 231-236 of the manuscript with Track Changes.

20. 193 what does "almost constant" mean in this context? Can you make this statement quantitative?

We calculate the coefficient of variation (cv = sigma/mean) to quantify the temporal variation. Results are presented in the manuscript as follows. "The standard deviation of bed elevation

becomes quite stable during the remaining conditioning phase, as well as during the hydrograph phase, despite the fact that degradation is evident as the flow approaches its peak value. For the standard deviation of bed elevation during the conditioning phase, we calculate the coefficient of variation (cv) for REF2 (15), which has the longest conditioning phase. The result shows a value of 0.038 from t = 15 minutes to the end of the conditioning flow. For the standard deviation of bed elevation during the hydrograph phase, we calculate the cv for all experiments. The results show that the values of cv vary between 0.031 and 0.075." See Lines 237-243 of the manuscript with Track Changes.

21. 198 please add in symbols to mark the time of the actual measurements.

We replot Figure 4 according to the suggestion of the reviewer.

22. 198 is it possible to add error bars to these plots?

For the standard deviation of bed elevation, we estimate the uncertainties as follows. We scanned the floor of the flume twice and calculated the standard deviations of the scanned DEM. The floor of the flume was horizontal and flat, with no sediment on the bed. Theoretically, the standard deviation of the DEM should be zero. Therefore, the calculated standard deviations of the flume floor could be regarded as an estimation of the uncertainty of the calculated values during the experiment.

The way we estimate the uncertainties is explained in Section 2 of the paper. See Lines 186-190 of the manuscript with Track Changes. According to this method, we estimate the uncertainties of the results in Figure 4 to be in the range of 1.6~2.5 mm, which is close to the vertical resolution of the laser scans (1 mm). We present our estimation in the caption of Figure 4.

23. 204 What does "relatively stable" mean in this context? How was stability assessed?

We calculate the coefficient of variation (cv = sigma/mean) to quantify the temporal variation of the variables. Results are given in the text. See Lines 256-258 of the manuscript with Track Changes.

24. 205 What does "relatively constant" mean in this context? How was constancy assessed?

Again, we calculate the coefficient of variation. Results are given in the text. See Lines 260-262 of the manuscript with Track Changes.

25. 206 Interpretation, to discussion.

Thanks for pointing this out. We now move this interpretation to the Discussion Section. See Lines 509-512 of the manuscript with Track Changes.

26. 209 Please mark the time of the measurements on the plots and add error bars.

We replotted Figure 5. Time of the measurements are marked, and range bars are added to estimate uncertainties.

27. 228 Fig. 6c shows presumably not a derivative, but a ratio of discrete changes. Please change the notation accordingly (for example, by using delta symbols). Please add error bars.

We do not think we are showing a ratio of discrete changes. We tried to quantify the trend by linear regression. Here is what we said in the paper, "Such intra-step variations of sediment transport rate are investigated in Fig. 6(c), with the x axis being the averaged sediment transport rate of each step $Q_{sa}$ and the y axis being $d(Qs/Q_{sa})/dt$. The value of $d(Qs/Q_{sa})/dt$ is estimated by linear regression."

As for the uncertainties of the light table method, please see our reply to Question 14.

28. 229 During each of the hydrograph steps, there seems to be decline of transport rates over time. This may be due to a transient adjustment of the bed to the changed hydraulic conditions. It does not seem to me here that an equilibrium is achieved. Can the different stages then be meaningfully compared? How would a transient adjustment affect the interpretations?

Yes, this is what we tried to quantify in Figure 6c. A negative value of discrete change denotes a declining trend, whereas a positive value denotes an increasing trend. We agree with the reviewer that these are transient adjustments due to changed water and sediment supply, and the equilibrium is not achieved. Actually, the decreasing/increasing trends exist just because the bed is not in equilibrium. We explain a bit more in the text. See Lines 325-326 of the manuscript with Track Changes. The sediment transport in different stages were compared in Figure 6b and in the text (Lines 299-308 of the manuscript with Track Changes).

Done.

30. 238 How did you assess whether it agrees? What does agreement mean in this context?

When we say "agree with", we mean that the adjustments of sediment transport and the adjustments of bed elevation show similar patterns during the hydrograph phase. In order to make our statement clear, we rewrite it as follows. "Such adjustments of sediment transport rate are consistent with the process of channel deformation shown in Fig. 3. That is, for both sediment transport and channel deformation, results of REF7 (0.25) deviate from other experiments in Step 1 (larger sediment transport rate and more degradation in REF7 (0.25)), but collapse with other experiments in the following three steps." See Lines 304-308 of the manuscript with Track Changes.

31. 246 Please explain how you detected trends and give corresponding statistics. The interpretation is a little difficult, since there are no error bars for the data.

The trends were detected based on linear regression, as we have stated in the text. See Line 311

of the manuscript with Track Changes. A negative value of $d(Q_s/Q_{sa})/dt$ corresponds to a decreasing trend, and a positive value of $d(Q_s/Q_{sa})/dt$ corresponds to an increasing trend.

32. 260 How did you arrive at this assessment? REF7 is around 50% larger than REF3! It would make sense to add error bars to the measurements and a statistical test to actually show that there is no difference.

Our previous statement might be misleading. What we want to express was that a longer duration of conditioning flow does not lead to a reduced sediment output during the subsequent hydrograph. We add a comparison of the data and also a calculation of the correlation coefficient, which helps support our idea. The text is rephrased as follows. "It can be seen that the effect of conditioning duration on the total sediment output during the entire hydrograph phase is not evident: a longer duration of conditioning flow does not necessarily lead to a smaller (or larger) sediment output. The largest sediment output occurs in REF7 (0.25), which is 55% larger than the sediment output in REF3 (10) which has the smallest output, but is about the same as (only 4% larger than) the sediment output in REF6 (15). We further calculate the correlation coefficient between the total sediment output and the duration of conditioning flow, and obtain a value of r = -0.14, indicating that there is almost no correlation between the two parameters." See Lines 330-336 of the manuscript with Track Changes.

As for the error bar, there is no error bar for the trap data. Data shown in Figure 7 are not calculated values, but are the material weighted in the sediment trap. Errors could be introduced due to the 0.25 mm mesh size of the tail box, the resolution of the scale, etc. However, we think these errors are negligible. The word "calculate" at the beginning of this paragraph could be misleading. We rephrase the text as "Sediment collected in the trap/tailbox at the flume outlet allows us to plot the total amount of sediment output during each step of the hydrograph." See Lines 327-328 of the manuscript with Track Changes.

The uncertainties of the trap data cannot be evaluated, since the data do not correspond to calculated values but instead correspond to the material weighted in the sediment trap. Please see our reply to Question 32 (the second paragraph).

Done.

35. 274 How did you arrive at this interpretation? Maybe this is just an effect of the scale of the plot? It would be good to add error bars to the data here and some suitable statistical test.

To better illustrate our idea, we quantitatively compare the data between different experiments. We also calculate the correlation coefficient *r* to estimate the effect of stress history on sediment transport. We rephrase the text. See Lines 343-364 of the manuscript with Track Changes.

As for the error bars, again we state that there is no error bar for the trap data. Please see our

reply to Questions 32 and 33.

36. 290-298 how did you establish significance and what does 'more significant' mean in this context / how was this assessed?

We use the coefficient of variation (cv = sigma/mean) to support our idea. The value of cv is calculated for both the conditioning phase (after $t = 10$ hour, the beginning of the conditioning sees a drop in $D_{l10}$ so that the cv is not appropriate to quantify the fluctuation) and the first two steps of the hydrograph phase. Results are presented in the text. See Lines 381-397 of the manuscript with Track Changes.

37. 300 Please give some indications of the uncertainties of these measurements.

In our reply to Question 14, we explain the uncertainties of light table measurements (both total transport rate and characteristic grain sizes) in detail. We do not repeat the explanation here. Please refer to our reply to Question 14.

38. 303-308 this needs more detail if it is relevant for the central message of the paper. If not, consider deleting it.

We put more detail concerning fractional sediment transport in the Supporting Information.

39. 304 The use of the term 'equal mobility' has become ambiguous. Originally, it meant that the grain size distribution of the transport material matches the grain size distribution of the material found on the bed. However, it is now often used to mean an equal threshold of motion for all grain sizes. It is unclear from the context here which meaning is intended.

In our study, "equal mobility" means the first and original definition. That is, the grain size distribution of the sediment load matches the grain size distribution of be sediment on bed surface. We add the definition of "equal mobility" in the text. See Line 404 of the manuscript with Track Changes. This definition is also added in the Supporting Information when sediment mobility is discussed.

40. 334 transport rate

Done.

41. 365 please add error estimates to these calculations.

We now estimate the uncertainty associated with the calculation of $\tau_c^*$. The methodology is explained in the text. See Line 466-476 of the manuscript with Track Changes.

For the $\tau_c^*$ values back-calculated with Equation (1) (i.e., Meyer-Peter and Muller type relation), the estimated uncertainties are presented in Table 1. For the $\tau_c^*$ values calculated with the Equation (5) (i.e., the equation of Lamb et al. (2008)), the uncertainty is less than ±1%, and is therefore not

presented in Table 2 (but is explained in the text).

**42. 388 predictions**

Done.

**43. 397 This would imply that the stronger trends should be seen in d90 rather than d50, right? How does this expectation compare to the data?**

The reviewer might misunderstand. With experiments, Masteller and Finnegan (2017) found that the most drastic changes during the conditioning flow are the reorientation of the highest protruding grains into nearby available pockets. Such a reduction in the number of highly protruding grains eventually leads to a reduction of sediment transport rate and a more stable bed surface. Masteller and Finnegan (2017) also reported that this reorganization of bed surface, however, does not lead to an evident change of bed surface GSD or surface topography standard deviation.

Therefore, we do not expect that such reorientation would be reflected in the variation of $D_{90}$ or $D_{50}$, as mentioned by the reviewer. We revise the text in the paper to make this more clear. See Lines 528-530 of the manuscript with Track Changes.

**44. 405 either 'e.g.' or 'etc.', not both**

We delete "etc.". Thanks for pointing this out!

**45. 416 …the conditioning flow was…**

Done.

**46. 416 In the present paper…**

Done.

**47. 427 consisted**

Done.

**48. 430 consisted**

Done.

**Referee #2**

1. This paper presents a series of flume experiments detailing the effect of conditioning flow duration on the sediment flux experienced during a subsequent flood wave. The paper in itself is interesting and would be of interest to the readers of ESurf but at the moment the language is not tight enough and the reader is left to try and fathom out the main take home messaged from each set of data analysis. For example, throughout the results there are phrases like 'increases notably', 'significant degradation' etc. but it is not clear whether or not the authors have undertaken statistical analysis to support their results. If they have then the outputs of those statistical analysis need to be reported. It is also unclear how reproducible the results are – again this needs to be reported. Therefore, this leaves the reader wondering how important the reported trends are, especially given the data is relatively noisy. I would also argue that the majority of the data analysis is relatively basic, which in itself is not a problem, but when coupled with the subjective language used, as discussed above, the main 'story' of the manuscript is lost. That said there are places where I think the analysis could be taken a lot further e.g. analysis of bed surface topography is not really mentioned despite DEMs being collected and mentioned in the methodology. Finally, I have a significant issue with section 4.1 and the back calculation of tau*c using MPM and Wong and Parker since this regression is designed for a small range of known tau*c. Irrespective of this I am not sure of the worth of scaling tau*c from that derived right at the beginning of the experiment – the bed state there is not representative of a true fluvial system, rather is representative of artificial conditions caused by screeding and high sediment transport rates caused by initial scour. So basing analysis off this seems odd and slightly misleading. In this section the authors also say that 'only the slope effect cannot explain the observed range of tau*c' – it is not clear what they mean and given this paper is about steep slope environments this needs careful clarification and expansion.

We appreciate the reviewer for the constructive comments. In the revision of the manuscript, we undertake statistical analysis to support our ideas. The uncertainties of the measurements are estimated. The variations and the correlations of the data are analyzed. Methodology is explained in detail in the text. See Lines 186-197 of the manuscript with Track Changes. Results are presented in reply to the relevant questions of both reviewers, as well as in the manuscript.

As for the reproducibility of the results, the reviewer raised an important issue. In a recently published paper, Church et al. (2020) drew attention to similar issue. Moreover, they distinguished three levels of "reproducibility": (1) "repetition" which repeats the program of observations in the same exercise; (2) "replication" which is duplication of observations using similar resources but in an independent program; (3) "reproduction" which is confirmation of scientific principles using different resources in an independent program. In our paper, the repetition of the experimental results is tested by repeating the conditioning phase with the longest duration (REF6 (15) and REF2 (15)). The two experiments show similar results in terms of standard deviation of bed elevation, GSD of bed surface, sediment transport rate, and GSD of sediment load. However, the reproduction of the experimental results, which requires independent tests undertaken using different materials and/or different conditions of measurement, and is more significant for advancing of the science according to Church et al. (2020), has not been tested in this paper. In this regard, more efforts are needed in future studies to test the reproducibility of the conclusions given in this paper. We discuss this issue in the manuscript. See Lines 562-570 of the manuscript with Track Changes.

The reviewer mentioned that the data reported in the paper was relatively noisy, and that the analysis were relatively basic. We add statistical analysis in the revised manuscript. Details of the statistical analysis are provided throughout this response file (see our reply to related comments of both reviewers). As for the analysis of the bed surface topography, we now calculate both the mean and the standard deviation of the DEM. We explain this in more detail in our answers to Questions 17 and 19 of Reviewer #2.

In the paper, when implementing the MPM relation modified by Wong and Parker (2005) for the estimation of $\tau_c^*$, we assumed that the MPM type relation holds under the condition of our experiments. It is worth mentioning that in a newly published paper, Hassan et al. (2020) applied three different methods to estimate the threshold of sediment motion in a gravel-bed river, including (1) back calculation with the Wong and Parker (2005) relation of MPM; (2) back calculation with the Wilcock and Crowe (2003) relation; and (3) the relation of Church et al. (1998). Estimation with the three different methods shows very similar temporal trend and variability (0.035~0.075 with Wong-Parker relation in their case), which implies that the specific function that is applied does not matter that much. We cite the work of Hassan et al. (2020) in the manuscript. See Lines 416-417 and 431-433 of the manuscript with Track Changes.

We would like to clarify that in the paper, $\tau_c^*$ is scaled against the value at $t$ = 15 minute, rather than at the very beginning of the conditioning phase. According to the experimental results (Figures 3, 4, and 5 in the manuscript), adjustments of the bed topography and surface texture have been accomplished within the first 15 minutes (and become rather insignificant after that). Therefore, we do not agree that "the bed state (as the base for scaling) is not representative of a true fluvial system, rather is representative of artificial conditions caused by screeding". Besides, the main purpose of the scaling is to facilitate the comparison of the temporal variation of $\tau_c^*$ among different experiments.

As for the slope effect, Reviewer #1 raised similar question. Please refer to our reply to Question 2 of Reviewer #1. More specifically, we say the following in the paper (Lines 462-464 of the manuscript with Track Changes). "Such discrepancies could be ascribed to the fact the relation of Lamb et al. (2008) considers only the influence of bed slope, but without considering the effects of other mechanisms like organization of surface texture, infiltration of fine particles, etc. These potential effects are discussed in more detail in Section 4.2."

2. I really like it when authors include an implications section in their papers and so it is good to see this included in the presented paper however some of the text in this section feels much more like discussion and framing of the authors results within the wider literature of which there was relatively little of in the paper up until that point. So, I think the discussion and implications sections of the paper need re-framing slightly so the discussion section properly frames the results within the wider literature and the implications section talks about the bigger picture and importance of the findings more broadly. If the authors do this, I think it will be much clearer to the reader about the new findings which this paper had generated – to make this even easier for the reader there are places where paragraphs could be re-structured such as to lead with the findings from this paper before framing within the previous literature. This will help make it crystal clear where the additional knowledge is. For these reasons I would suggest a rejection of the paper with a strong encouragement to resubmit once the issues have been addressed. Line by line changes are also suggested below to help the authors revise their manuscript.

The Discussion Section has been rephrased according to the related comments from both reviewers. We really appreciate this!

3. Line 24 – arguably not just mountain streams.

We delete the word "mountain".

4. Line 26 – I would consider adding Masteller 2019 JGR paper in here as it seems relevant.

We add this reference here.

5. Line 28 – what do you mean by average flow regime?

When we say "average flow regime", we mean that most sediment transport relations for mountain streams are based on constant flow, which may be regarded as an average of the high unsteady flow regime of mountain streams.

6. Line 44 – remove etc in the citations.

Done.

7. Line 52 – should be Haynes

Thanks for pointing this out! We have corrected the typo.

8. Line 82 – from reading your methodology I would argue that you don't run experiments which consist of 'extended cycles' – for me that reads as you cycled hydrographs but this is not what you did – instead you changed the length of a period of conditioning flow before exposing that bed to the rising limb of the hydrograph.

We agree with the reviewer. We revise the text to avoid misunderstanding. The text now reads "In this paper, flume experiments consisting of high and low flow are conducted to study this problem". See Line 89 of the manuscript with Track Changes.

9. Line 88 – can you be more explicit with what you mean by guided by?

Here we mean that the East Creek is the prototype for the experimental arrangements in this paper. This was explained in more detail in the subsequent text, including the design of flow discharge (Lines 116-119 of the manuscript with Track Changes) and sediment grain size distribution (Lines 132-135 of the manuscript with Track Changes).

10. Line 89 – to investigate the study objective……

Done.

11. Line 104-106 – so was the flow rate directly scaled?

Yes, the flow rate was directly scaled as written in the paper.

12. Line 105 – I don't think that you ran a hydrograph – you ran a rising limb of a hydrograph but not a full hydrograph. The results you would have come up with by running a full hydrograph would have been very different to those which you report here

The reviewer is correct. To avoid misunderstanding, we add the following sentence in the manuscript. "It should be noted that in the experiments, we only implemented the rising limb of the hydrograph/sedimentograph, rather than a full hydrograph/sedimentograph with both rising and falling limbs. Rather than studying river adjustment during a flow hydrograph, we aimed at determining the influence of conditioning time on bedload and bed surface arrangements as flow rates increased." See Lines 102-105 of the manuscript with Track Changes. The reason that we implemented only the rising limb is that we would like to focus on how the stress history effect is influenced with the increase of flow intensity.

13. Line 108- 111 – more justification and reasoning is needed in this section – what are the details of the trail experiments? How exactly were the feed rates chosen?

We rephrase the text as "For each step of the hydrograph, the feed rate of sediment was specified to be close to the transport capacity of the flow. Determination of the sediment supply rates was facilitated by a numerical model which was calibrated for similar experimental conditions (Ferrer-Boix and Hassan, 2014)." See Lines 120-122 of the manuscript with Track Changes.

14. Line 117 – why was this sediment scaling chosen?

The scaling factor used in our experiments (1:4) is similar to that used by Chartrand et al. (2018) (which is 1:5). Also, the finest fractions (that less than 0.5 mm) were excluded to avoid sediment transported in suspension.

15. Line 123 – what impact does feeding the bulk sediment rather than feeding a GSD which is representative of the transported sediment have on the surface development?

We compared the GSD of the bulk sediment against the GSDs of bedload measured with the trap. Results are shown in the subsequent figure. It can be seen that the GSDs of bedload vary over a relatively wide range according to the flow intensity, sediment supply rate, bed surface texture, etc. The GSD of bulk sediment falls in about the middle of this range. Therefore, we think that the bulk sediment is a good representative of the bedload in average. Moreover, for gravel-bed rivers, the bulk material is often regarded as a representative of the sediment supply texture in the long term.

[Figure]

16. Line 127 – 129 – how often did you measure the bed surface profile (It is difficult to tell from fig 1) – was this just down the channel centre line?

The measurement of bed surface profile was denoted as "DEM" in Figure 1. Information related to the measurement frequency was given in the text. See Lines 177-180 of the manuscript with Track Changes. The DEM of the bed surface was measured for the whole channel bed, rather than just the channel centerline.

17. Line 131- 136 and Line 156 - What impact did draining the flume have on the transport dynamics – did you assess this? How were the DEMs detrended? You need a sentence or more in here justifying why you are using std of elevations as a measure of surface topography.

To minimize the influence of draining the flume, the flow was lowered slowly with the time interval between the stop of the pump and the stop of the flow being about 3 to 4 minutes, and the subsequent measurements (point gauge, light table, etc.) were taken after the flow was back to stable. We revised the text accordingly to present more details. See Lines 180-183 of the manuscript with Track Changes.

The DEMs were detrended based on linear regression. We add the information in the text. See Line 151 of the manuscript with Track Changes.

We used the standard deviation of elevation as a measure of surface topography, since our recently published paper (Chen et al., 2020) showed that the standard deviation of bed elevation is a good descriptor for bed roughness of gravel-bed rivers. Reviewer #1 asked a similar question. We explain this in the manuscript (Lines 231-233 of the manuscript with Track Changes).

18. Line 148 – was the shear stress corrected for side walls? If so how?

A side wall correction is not implemented in this paper. In our experiments, the flow depth varies in the range of 0.046-0.086 m, which leads to a width/depth ratio of 6.4-12.0. According to Julien (2010), side wall effects are not significant when the width/depth ration is larger than 5.

We apply a side wall correction for the two cases with largest and smallest depth, using the method of Chiew and Parker (1994). Results show that, for the case of the smallest depth, the shear

stress without side wall correction is 15.7 Pa, and the shear stress with side wall correction is 15.2 Pa (a reduction of 3.0%). Whereas for the case of the largest depth, shear stress without a side wall correction is 25.3 Pa, and shear stress with side wall correction is 24.7 Pa (a reduction of 2.7%). Therefore, we think it is reasonable to neglect side wall effects in the analysis.

19. Line 166 – I would have thought it would have been useful to present the statistical moment analysis of the bed scans so you can properly link the development of the bed surface with some of the sediment dynamics. I think it would also be useful to plot the surface DEM evolution to allow readers to better appreciate and understand how the surface evolves.

We have calculated the standard deviation (second order moment) of the DEM, as shown in Figure 4. We chose the standard deviation because our recent research (Chen et al., 2020) found it to be a good descriptor for bed roughness of gravel-bed rivers. We also studied the mean (first order moment) of the DEM. This is analyzed in terms of the mean difference of bed elevation during each flow stage, which represents the overall channel aggradation/degradation in each stage. Results are presented in Table 1 ($\Delta z_b$) as well as in the text associated with Figure 3.

The DEMs of bed surface at different times during the experiment are now plotted in the Supporting Information (Section S1).

20. Line 168 – I am not sure exactly what you mean by longitudinal DEM – why average it over the cross section? A lot of the previous literature on stress history which has undertaken DEM analysis has shown that the surface develops significant spatial complexity which will be lost by the averaging you have undertaken

Here we averaged the DEMs over the cross section with the purpose to study the overall aggradation/degradation in the longitudinal profile during different stages of the experiments. We add the following sentence in the paper. "The DEM over the cross section is used here to study the overall aggradation/degradation of the channel." See Lines 207-208 of the manuscript with Track Changes.

21. Line 180 – what do you mean by further analysis of the DEM? Where is this analysis?

Our determination of the bedform is visually based on the DEM as well as the direct observation of the channel bed. We realized that the statement was not accurate without further quantification. Therefore, we have removed the sentence in the text to avoid misunderstanding. See Lines 223-224 of the manuscript with Track Changes. We also suggest the reviewer to refer to the DEM that we add in the Supporting Information (Section S1).

22. Line 205 – should be noting not noted

Done.

23. Line 205 – what do you mean by keeps relatively flat?

We mean that the characteristic grain sizes of bed surface do not change much with time. To support our idea, we add statistical analysis by calculating the coefficient of variation (cv = sigma/mean). Results are given in the text. See Lines 256-262 of the manuscript with Track Changes.

24. Line 212 – how accurate is the light table?

Reviewer #1 asked a similar question. We have answered in detail and thus do not repeat here. Please see our reply to Question 14 of Reviewer #1. Thanks!

25. Line 216 – 217 – again be specific – what do you mean by very large? Gradually dropping? Small and relatively constant?

We add some analysis. The following sentences are added in the text.
"In the first 15 minutes, the sediment transport rates drop from more than 500 kg/h to less than 100 kg/h. Afterwards, it takes about another 2 hours for the sediment transport rates to drop to close to 1 kg/h."
"For REF2 (15) and REF6 (15) which have the longest conditioning phase, the sediment transport rates between t = 8 hour and the end of conditioning phase (t = 15 hour) show mean values of 0.35 kg/h (standard deviation = 0.22 kg/h) and 0.37 kg/h (standard deviation = 0.24 kg/h), respectively."
See Lines 279-285 of the manuscript with Track Changes.

26. Line 223 - why is this analysis in the supplemental information? The analysis you have undertaken up to this point in the paper would really seem to benefit from this further analysis

We put this material in the Supporting Information as we think it does not belong to the main conclusions of this paper. Besides, the current manuscript is already very long.

27. Line 268 and 275 – again be specific and give the statistical outputs if you have undertaken statistical analysis.

We add a statistical analysis in the text. See Lines 344-346 and Lines 354-357 of the manuscript with Track Changes.

28. Line 279 – I am not sure I agree that the five experiments show similar sediment outputs – there may not be a systematic trend related to condition flow duration but there are certainly differences between them.

We rephrase the text as follows. "In the last step of the hydrograph, with the flow discharge and sediment supply approaching their peaks, the difference in sediment output among the five experiments again becomes small, with the values ranging between 72.1 kg in REF4 (2) and 119.6 kg in REF6 (15). This demonstrates that little influence of stress history remains in this step." See Lines 361-364 of the manuscript with Track Changes.

29. Line 303 – 308 – again why is this data in the supplemental information – I would have thought it would have been a really important addition to your paper and provided a lot of useful context from which to hang your discussion

We thank the reviewer for the positive comment. We put this material in the Supporting Information as we think it does not belong in the main conclusions of this paper. Besides, the current manuscript is already very long.

30. Line 334 – sediment transport rate

Done.

31. Line 337 – what do you mean by basically show an increasing trend – be specific

We mean that the value of dimensionless sediment transport rate $q^*$ increases with the increase of Shields number $\tau^*_{s50}$. Our calculation shows a rather good correlation between $\tau_{s50}{}^*$ and $\log(q_s{}^*)$ (consistent with the semi-log scale of Figure 9(a)), with the correlation coefficient being 0.58. We add this result in the text to support our conclusion. See Lines 440-441 of the manuscript with Track Changes.

32. Line 376 – remove etc. from the citations

Done.

33. Line 391 – what implications – can you be specific?

We remove this sentence.

34. Line 405–413 – I am afraid I don't see the relevance of this paragraph to the paper.

We would like to keep this paragraph. The idea of this paragraph is that, in our flume experiments sediment was fed at the upstream with low supply during conditioning flow and high supply during flood. However, natural mountain streams are more complicated and not always like this. Sediment supply during low flow could be abundant (e.g. in the form of landslides), especially at places where the hillslopes are active. Such effects are not included in our experiments, but are discussed in this section. We think this would also merit future research.

35. Line 415 – more should have been made in the discussion of the comparison between the work presented in the current paper and the results of Haynes and Pender (2007) since this is a very relevant study which would have provided really useful comparators.

We add more discussion in the text. See Lines 544-557 of the manuscript with Track Changes.

36. Line 419 – again I am not sure I agree with you here – the data presented in this paper has shown

that the effects of stress history are effectively cancelled out under higher flows in any subsequent flood. However, in line 419 you say 'might be more lasting during subsequent flood' – this seems counter to the rest of the message in the paper.

We rephrase the text to avoid misunderstanding. See Lines 544-557 of the manuscript with Track Changes. We appreciate the reviewer for the comment.

References

Chartrand, S. M., Jellinek, A. M., Hassan, M. A., and Ferrer-Boix, C.: Morphodynamics of a width-variable gravel bed stream: New insights on pool-riffle formation from physical experiments, Journal of Geophysical Research-Earth Surface, 123(11), 2735-2766, https://doi.org/10.1029/2017JF004533, 2018.

Chen, X., Hassan, M. A., An, C., and Fu, X.: Rough correlations: Meta-analysis of roughness measures in gravel bed rivers, Water Resources Research, 56, e2020WR027079, https://doi.org/10.1029/2020WR027079, 2020.

Chiew, Y.-M., and Parker, G.: Incipient sediment motion on non-horizontal slopes, Journal of Hydraulic Research, 32(5), 649-660, https://doi.org/10.1080/00221689409498706, 1994.

Church, M., Dudill, A., Venditti, J. G., and Frey, P.: Are Results in Geomorphology Reproducible? Journal of Geophysical Research-Earth Surface, 125(8), e2020JF005553, https://doi.org/10.1029/2020JF005553, 2020.

Church, M., Hassan, M. A., and Wolcott, J. F.: Stabilizing self-organized structures in gravel-bed stream channels: field and experimental observations, Water Resources Research, 34, 3169-3179, 1998.

Curran, J. C., and Wilcock, P. R.: Effect of sand supply on transport rates in a gravel-bed channel, Journal of Hydraulic Engineering, 131(11), 961–967, https://doi.org/10.1061/(ASCE)0733-9429(2005)131:11(961), 2005.

Hassan, M. A., Saletti, M., Johnson, J. P. L., Ferrer-Boix, C., Venditti, J. G., and Church, M.: Experimental insights into the threshold of motion in alluvial channels: sediment supply and streambed state, Journal of Geophysical Research-Earth Surface, e2020JF005736, https://doi.org/10.1029/2020JF005736, 2020.

Julien, P. Y.: Erosion and sedimentation, Cambridge University Press, New York (USA), 390 p, 2010.

Lamb, M. P., Dietrich, W. E., and Venditti, J. G.: Is the critical Shields stress for incipient sediment motion dependent on channel-bed slope? Journal of Geophysical Research-Earth Surface, 113, F02008, doi:10.1029/2007JF000831, 2008.

Masteller, C. C., and Finnegan, N. J.: Interplay between grain protrusion and sediment entrainment in an experimental flume, Journal of Geophysical Research-Earth Surface, 122, 274–289, https://doi.org/10.1002/2017GL076747, 2017.

Venditti, J. G., Dietrich, W. E., Nelson, P. A., Wydzga, M. A., Fadde, J., and Sklar, L.: Mobilization of coarse surface layers in gravel-bedded rivers by finer gravel bed load, Water Resources Research, 46, W07506, https://doi.org/10.1029/2009WR008329, 2010.

Wilcock, P. R., Kenworthy, S. T., and Crowe, J. C.: Experimental study of the transport of mixed sand and gravel, Water Resources Research, 37(12), 3349–3358, 2001.

Wong, M., and Parker, G.: Reanalysis and correction of bed-load relation of Meyer-Peter and Müller using their own database, Journal of Hydraulic Engineering-ASCE, 132, 1159–1168, doi:10.1061/(ASCE)0733-9429(2006)132:11(1159), 2006.

Zimmermann, A., Church, M., and Hassan, M. A.: Video-based gravel transport measurements with a flume mounted light table, Earth Surface Processes and Landforms, 33(14), 2285-2296, https://doi.org/10.1002/esp.1675, 2008.